# Sex-dependent rescue of memory and synaptic deficits in AD model mice by increasing PSD-95 palmitoylation
Yixing Du[1], Katie Prinkey[1], Andrew Q. Pham[1,4], Amber Lawrence[1,4], Celeste Morales[1,4], Maureen Dinata[1,4], Marlenne Gutierrez[1,4], Ahmed Khalil [1,4], Medha Sharma[2], Robert A. Rissman[3], Mehreen Manikkoth[1], Ian Baick[1], Haritha Karthikeyan[1] & Kim Dore [1] ✉

PSD-95, a major scaffolding protein, requires palmitoylation to remain at synapses where it plays critical roles in synaptic structure and function. Here, we show that PSD-95 palmitoylation is specifically reduced in the hippocampus of female Alzheimer's disease (AD) model mice. Accordingly, these mice have significant memory deficits that are not observed in male AD model mice. Systemic injections of Palmostatin B, a depalmitoylating enzyme inhibitor (including the one acting on PSD-95), rescues memory deficits in female AD model mice and restores PSD-95 palmitoylation levels. Importantly, both synaptic structure and function are impaired in female AD model mice, and these deficits are normalized in Palmostatin B injected animals. This drug has no effects on amyloid plaques or GFAP levels, indicating that the rescue of behavioral and synaptic deficits is not due to effects on plaque or astrogliosis related AD pathology. Our data instead suggest that the sex-dependent rescue we observe is mediated by the stabilization of small, vulnerable dendritic spines. This study demonstrates that increasing PSD-95 palmitoylation might be an effective way to protect synapses from AD pathology and therefore a promising therapy for AD.

One of the most perplexing features of Alzheimer's disease (AD) is the fact that women have a significantly higher risk of developing the disease[1–4], which cannot be explained by their higher life expectancy as previously thought[4]. Women consistently show earlier and faster hippocampal atrophy then men[2,3] despite having comparable levels of pathology[5]. Recent research looking into the possible causes of this sex difference suggests that hormones, metabolism and inflammation are most likely involved[1]. In AD mouse models, female mice also perform significantly worse than male mice in memory-related behavioral tests, indicating that transgenic mice can replicate this characteristic of AD[6–8]. Interestingly, sex differences in the size and density of dendritic spines in the hippocampus were recently reported[9,10]. Synaptic loss has been proposed as the initial target in AD, correlating far better with cognitive symptoms at all stages of the disease than other pathological features[11–13]. Therefore, sexual dimorphism in synaptic vulnerability might be an important factor mediating sex differences in AD. Moreover, despite growing insights into synaptic loss as a hallmark of AD, the underlying molecular pathways remain elusive, leaving a critical gap in therapeutic strategies that directly address this detrimental

process. PSD-95, a major postsynaptic scaffolding protein, is significantly depleted in the brains of AD patients[14], in several AD mouse models[15–17], as well as in neurons exposed to amyloid-beta (Aβ)[18]. Indeed, PSD-95 is reduced before pre-synaptic proteins[19] and in individuals with mild cognitive impairment[20], suggesting that PSD-95 reduction is an early event in disease progression.

The amount of synaptic PSD-95 is controlled by dynamic S-palmitoylation, which mediates the insertion of PSD-95 in postsynaptic membranes[21]. S-palmitoylation is a posttranslational modification in which palmitic acid is added to a cysteine residue via a thioester bond. In contrast with other lipid modifications, such as N-palmitoylation, O-palmitoylation, myristylation and prenylation, S-palmitoylation is reversible[22], permitting therapeutic intervention on this process. While palmitic acid is the main substrate for S-palmitoylation, other long-chain fatty acids like stearic acid can also participate in S-palmitoylation[23]. Consequently, the field is shifting toward using the more general term S-acylation, as current detection assays cannot distinguish which type of fatty acid has been added. However, to remain consistent with previous literature on PSD-95 palmitoylation[21,24], we will

[1]Department of Neurosciences, University of California San Diego, La Jolla, CA, USA. [2]Department of Pharmacology, University of California San Diego, La Jolla, CA, USA. [3]Alzheimer's Therapeutic Research Institute, Keck School of Medicine, University of Southern California, San Diego, CA, USA. [4]These authors contributed equally: Andrew Q. Pham, Amber Lawrence, Celeste Morales, Maureen Dinata, Marlenne Gutierrez, Ahmed Khalil. ✉e-mail: kdore@ucsd.edu

continue to use this term to describe S-acylation. Proper synaptic function depends on the precise localization of thousands of proteins, a process in which palmitoylation, affecting 40–50% of synaptic proteins[25,26], is thought to be critically involved. PSD-95 palmitoylation is essential for its synaptic clustering as well as that of glutamate receptors[27–30]. Moreover, long-term synaptic depression is associated with reduced PSD-95 palmitoylation and its subsequent removal from synapses[31], while homeostatic scaling resulted in increased PSD-95 palmitoylation[32]. Therefore, a decrease of PSD-95 palmitoylation might be an unknown step leading to synaptic loss during AD. In a recent study[33] looking into palmitoylation-associated genes using bioinformatics analysis methods, palmitoylating enzyme genes were reduced in AD patients while depalmitoylating enzymes gene were increased; suggesting a significant reduction of protein palmitoylation in AD patients and that interventions increasing palmitoylation could be beneficial.

Acyl-protein thioesterases or "depalmitoylating enzymes" are responsible for the depalmitoylation of proteins[22], and alpha/beta hydrolase domain-containing protein 17 isoforms (ABHD17a, ABHD17b, and ABHD17c) were identified as the most efficient enzymes towards PSD-95[24]. Palmostatin B (Palm B) is a potent inhibitor of those enzymes[34] and was shown to increase the intensity and size of PSD-95 clusters in cultured hippocampal neurons[30]. We discovered that Palm B treatment in vitro can rescue Aβ-induced synaptic depression and Aβ-mediated effects on dendritic spines[35]. Moreover, Palm B was recently effectively used in vivo as a protective therapy against melanoma[36] at a concentration of 10 mg/kg without any apparent toxicity. Importantly, while most of the palmitoylated proteome is stably palmitoylated[37], PSD-95 is among the proteins most rapidly palmitoylated/depalmitoylated[24]; thus, low concentrations of a systemically delivered depalmitoylation inhibitor would be expected to have a strong effect on levels of palmitoylated PSD-95.

Here we show that palmitoylation of both total protein and PSD-95 are significantly reduced in the hippocampus of 9–10-month-old female AD model mice while no such effect was observed in male mice. Palm B injections in 9–10-month-old female AD model mice rescued memory deficits observed in the Morris water maze test and rescued levels of palmitoylated PSD-95, without affecting total palmitoylation. This indicates that IP injections of this drug can access brain synapses in vivo and suggests that Palm B mostly affects PSD-95 palmitoylation. Interestingly, male mice did not show significant memory deficits, and Palm B had no effect on their performance or protein palmitoylation. This indicates that levels of palmitoylated PSD-95 are correlated with spatial memory. We saw no differences in Aβ plaques or GFAP signal between vehicle or Palm B-treated AD model mice of both sexes, suggesting that the rescue of hippocampal-dependent memory is not due to a direct effect of Palm B on plaque or astrogliosis related AD pathology. Moreover, electrophysiological recordings of miniature excitatory synaptic currents (mEPSCs) revealed that synaptic transmission deficits in female AD model mice are restored in Palm B-injected animals. This rescue of synaptic function was also seen at the structural level, as Palm B restored dendritic spine density and morphology in female AD model mice. Since deficits start to appear as early as 4–6 months in this mouse model[38,39], these results in older animals suggest that our approach can rescue deficits in animals with significant AD pathology. Thus, instead of acting on AD pathology, enhancing PSD-95 palmitoylation with Palm B can reverse memory deficits, synaptic dysfunction, and dendritic spine pathologies in the hippocampus of older, symptomatic female AD model mice. Consequently, PSD-95 depalmitoylating enzyme appears to be an exciting new drug target for AD.

## Results
### Total protein palmitoylation and PSD-95 palmitoylation are reduced in the hippocampus of female APP/PS1 mice

Since a large proportion of synaptic proteins are palmitoylated[25,26] and that palmitoylation is expected to be reduced in AD[33], we aimed to determine if total protein palmitoylation was altered in the hippocampus of 9–10-month-old male and female APP/PS1 mice. Using Western blotting with an antibody detecting all palmitoylated proteins (see "Methods"), we found

that total protein palmitoylation was reduced in female APP/PS1 mice: $1.0 \pm 0.4$ for WT mice, $N = 6$; vs. $0.25 \pm 0.05$ for APP/PS1 mice, $N = 6$; $p < 0.01$ (Fig. 1A, B). No such reduction was observed in male mice, suggesting that palmitoylation of hippocampal proteins might be affected in female APP/PS1 mice only. Dendritic spines are an early target in the APP/PS1 AD mouse model[40–43]; therefore, reductions in synaptic proteins are expected. To evaluate alterations in PSD-95 levels in APP/PS1 mice, we quantified total PSD-95 in these lysates. We found that both male and female APP/PS1 mice had PSD-95 levels that were similar to WT mice in hippocampal lysates (Fig. 1E–H). While some publications reported lower PSD-95 levels in 7–8 months old APP/PS1 mice[15,43], PSD-95 levels were higher in 8-week-old APP/PS1 mice[44], suggesting that levels of PSD-95 can change during disease progression in APP/PS1 mice. PSD-95 palmitoylation is essential for its synaptic clustering[27] and to protect synapses from Aβ[35]. Moreover, PSD-95 palmitoylation was found to be increased during synaptic scaling[32], suggesting that PSD-95 palmitoylation is important for this process. We thus characterized PSD-95 palmitoylation in APP/PS1 mice using the APEGS assay, a biochemical assay developed by the Fukata group[45] that can simultaneously measure levels of non-palmitoylated PSD-95 and PSD-95 palmitoylated at one or two sites. We found that PSD-95 palmitoylation, as quantified using a weighted ratio (Palm ratio, see "Methods"), was significantly reduced in female APP/PS1 mice: $0.69 \pm 0.04$ for WT mice, $N = 5$; vs. $0.51 \pm 0.03$ for APP/PS1 mice, $N = 8$; $p < 0.01$ (Fig. 1I, J). Interestingly, this reduction in the Palm ratio is mediated by a significant increase in non-palmitoylated PSD-95 and a decrease in PSD-95 palmitoylated at 2 sites (Supplementary Fig. 1). The amount of singly palmitoylated PSD-95 did not differ between WT and APP/PS1 female mice, suggesting that this PSD-95 population is stable and do not contribute to the observed changes. Moreover, the Palm ratio was not affected in male mice: $0.80 \pm 0.03$ for WT mice, $N = 5$; vs. $0.82 \pm 0.06$ for APP/PS1 mice, $N = 6$; $p = 0.76$ (Fig. 1K, L) and no differences in the relative amounts of non-palmitoylated or PSD-95 palmitoylated at one or two sites were observed (Supplementary Fig. 1). To test if this reduction in PSD-95 palmitoylation was specific to the hippocampus, we performed the APEGS assay in frontal cortex lysates and no reduction was detected in the Palm ratio in female APP/PS1 mice (Supplementary Fig. 2), indicating that PSD-95 palmitoylation is specifically reduced in the hippocampus in this AD mouse model.

Immunohistochemistry (IHC) experiments in fixed brain slices from the same animals as for the biochemical characterization (Fig. 1) were also performed to assess palmitoylated PSD-95 levels with a different approach. Using a commercial antibody designed to detect palmitoylated PSD-95 named PF11, and antibodies against PSD-95 and MAP2 (microtubule-associated protein 2, identifying dendrites), we quantified PSD-95 and PF11 levels in the *Stratum radiatum* of the hippocampal CA1 region in WT and APP/PS1 mice (see "Methods"). Even after optimization of the IHC conditions using PSD-95 knock-out (KO) mice, residual staining is apparent in the PSD-95 KO mice when probed with the PF11 antibody (Supplementary Fig. 3A). It is also important to note that there is a lack of co-localization between the PSD-95 antibody and the PF11 antibody (Supplementary Figs. 3A; 4; 5; 6C, D and 9). These data suggest that the PF11 antibody produces non-specific signal. As such these figures are included only as Supplementary Figs. They may be considered indicative, but should not be interpreted as confirmatory of the APEGS data. Similarly to what we observed with the APEGS assay, we saw a significant decrease in PF11 signal intensity in female APP/PS1 mice and no change in male mice (Supplementary Fig. 4A–C). PSD-95 levels were also not significantly different between WT and APP/PS1 mice for both sexes (Supplementary Fig. 4C). Moreover, we found that both PSD-95 and PF11 signals were decreased in the subiculum sub region of the hippocampus in female APP/PS1 mice but not in males or in the dentate gyrus (Supplementary Fig. 5), suggesting that the subiculum is particularly vulnerable to AD pathology.

To further evaluate sex differences, we quantified hippocampal PSD-95 using Western blotting and found that while WT mice had similar PSD-95 levels, female APP/PS1 mice had significantly lower PSD-95 amounts as compared to male APP/PS1 mice (Supplementary Fig. 6A, B). Similar results

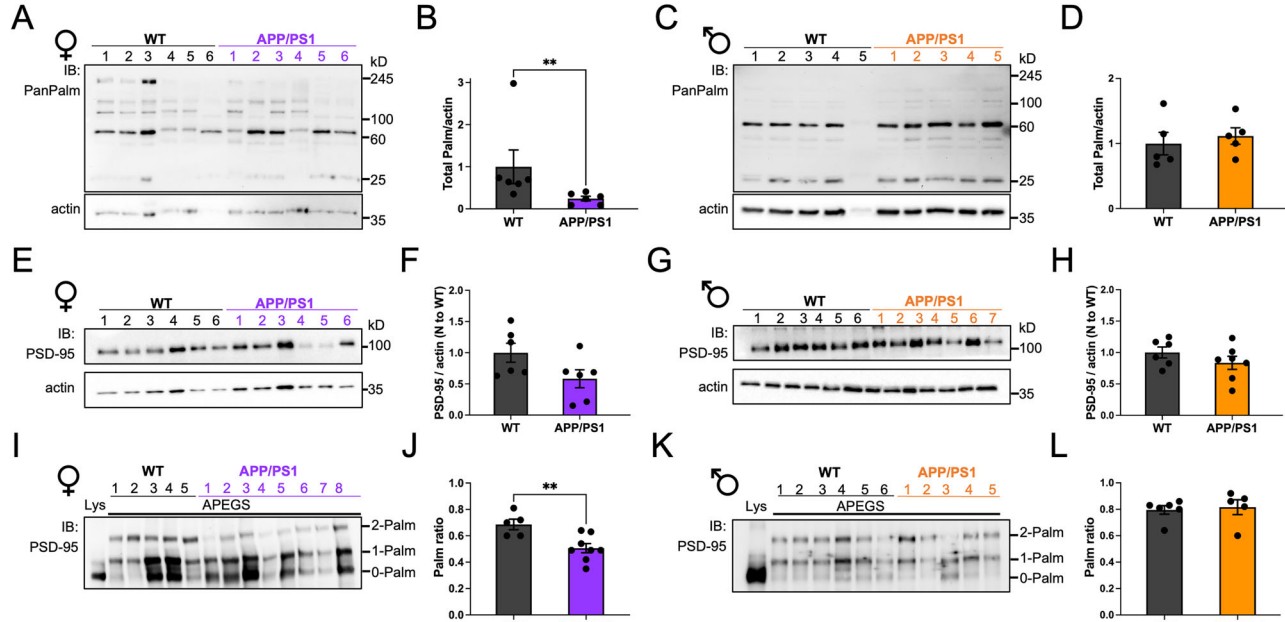

**Fig. 1 | Total protein palmitoylation and PSD-95 palmitoylation are reduced in the hippocampus of female APP/PS1 mice. A** Western blot of total protein palmitoylation in hippocampal lysates from female mice, each lane is a different animal. Immunoblotting with PanPalm antibody (see "Methods") and actin. See also Supplementary Figs. 1 and 6. **B** Quantification of the Western blot shown in (**A**), $N = 6$ WT and APP/PS1 mice. ** $p < 0.01$, Mann–Whitney test. **C** Western blot of total protein palmitoylation in hippocampal lysates from male mice. **D** Quantification of the Western blot shown in (**C**), $N = 5$ WT and APP/PS1 mice. non-significant, Mann–Whitney test. **E** Western blots of hippocampal lysates from female mice. Immunoblotting with PSD-95 and actin. See also Supplementary Figs. 1 and 6. **F** Quantification of PSD-95 hippocampal levels in the Western blots shown in (**E**).

$N = 6$ WT and APP/PS1 mice. **G** Western blots of hippocampal lysates from male mice. Immunoblotting with PSD-95 and actin. See also Supplementary Figs. 1 and 6. **H** Quantification of PSD-95 hippocampal levels in the Western blots shown in (**G**). $N = 6$ WT and $N = 7$ APP/PS1 mice. **I** Western blot of female hippocampal samples that underwent the APEGS assay. See also Supplementary Figs. 1 and 2. **J** Quantification of the Palm ratio in the Western blot shown in (**I**), $N = 5$ WT and $N = 8$ APP/PS1 mice. ** $p < 0.01$, Mann–Whitney test. **K** Western blot of male hippocampal samples that underwent the APEGS assay. **L** Quantification of the Palm ratio in the Western blot shown in (**K**), $N = 6$ WT and $N = 5$ APP/PS1 mice. Not significant, Mann–Whitney test. Error bars in all panels represent ± SEM.

were obtained by measuring PF11 and PSD-95 levels using IHC (Supplementary Fig. 6C, D). These results indicate that female APP/PS1 mice have less hippocampal PSD-95 than male APP/PS1 mice, which might be correlated with the stronger AD phenotype observed in female APP/PS1 mice[6–8,46–48].

## Palm B rescues memory deficits and PSD-95 palmitoylation in APP/PS1 female mice

In contrast with other synaptic proteins, PSD-95 palmitoylation is very dynamic[24], suggesting that even a weak block of depalmitoylation could increase PSD-95 palmitoylation levels. Moreover, in our previous in vitro study Palm B had no effects in slices made from PSD-95 KO mice, indicating that PSD-95 is required for Palm B to affect synaptic transmission[35]. Therefore, in vivo injections of a low dose of Palm B should mainly affect PSD-95 palmitoylation. We thus treated 9–10-month-old male and female WT and APP/PS1 mice with 5 mg/kg Palm B or vehicle and tested these animals in the Morris water maze. Animals were injected 4 days prior to the beginning of training and received injections every 48 h throughout behavioral testing (Fig. 2A). This treatment plan was adapted from ref. 36 and takes into account the fast turnover of PSD-95 palmitoylation[24] and the intent to limit Palm B dosage. No differences between groups were observed during visible or invisible learning (Supplementary Fig. 7 and see "Methods"). Similarly, to other studies investigating spatial memory in mice[49,50], we used both a visible learning phase to provide an easier task for the mice and to rule out potential motor or vision deficits and an invisible-platform phase to directly assess spatial memory. Significant impairments were observed for female APP/PS1 mice during the probe test. This test is performed at the end of the Morris water maze protocol and consists in tracking mice in the pool without the platform present (representative trajectories for each experimental group are shown in Fig. 2B, C). Female APP/PS1 mice injected with

vehicle spent significantly less time in the quadrant where the platform was located (see Supplementary Fig. 7 for data in all quadrants): 46 ± 3% for WT mice, $N = 10$; vs. 29 ± 4% for APP/PS1 mice, $N = 14$; $p < 0.01$ (Fig. 2D), indicating a deficit in hippocampal-dependent memory. This deficit was rescued in Palm B injected APP/PS1 female mice: 29 ± 4% for vehicle injected mice, $N = 14$; vs. 41 ± 4% for Palm B injected mice, $N = 15$; $p < 0.05$ (Fig. 2D). Surprisingly, male APP/PS1 mice did not present significant deficits and Palm B injected animals had similar performance than the ones injected with vehicle: 44 ± 4% for vehicle injected mice, $N = 10$; vs. 43 ± 4% for Palm B injected mice, $N = 10$; $p = 0.36$ (Fig. 2E). It is known that male APP/PS1 mice have less AD pathology[6,46–48], which may explain better performance in behavioral tests such as the Morris water maze. Moreover, other studies also found that only female APP/PS1 mice had significant deficits in the Morris water maze test[6–8].

To determine if this Palm B-dependent rescue of memory deficits in female APP/PS1 mice coincides with an increase in PSD-95 palmitoylation, we characterized PSD-95 levels and palmitoylation state in the brains of animals that underwent behavioral testing. We found that Palm B did not affect total PSD-95 levels but increased the Palm ratio in female APP/PS1 mice: 0.38 ± 0.04 for vehicle injected mice, $N = 5$; vs. 0.59 ± 0.03 for Palm B-injected mice, $N = 5$; $p < 0.01$ (Fig. 2F–I). Moreover, we found that the increase in the Palm ratio is mediated by a decrease in non-palmitoylated PSD-95 and an increase in PSD-95 palmitoylated at 2 sites (Supplementary Fig. 8). Using IHC, we observed that the PF11 signal was significantly increased in female APP/PS1 mice injected with Palm B (Supplementary Fig. 9A, C) but not in males (Supplementary Fig. 9B, D). These results show that Palm B only affects PSD-95 palmitoylation in female APP/PS1 mice, which had lower amounts of palmitoylated PSD-95 to begin with (Fig. 1), suggesting that Palm B preferentially acts on weakened synapses. Because Palm B also inhibit other depalmitoylating enzymes, like acyl-protein

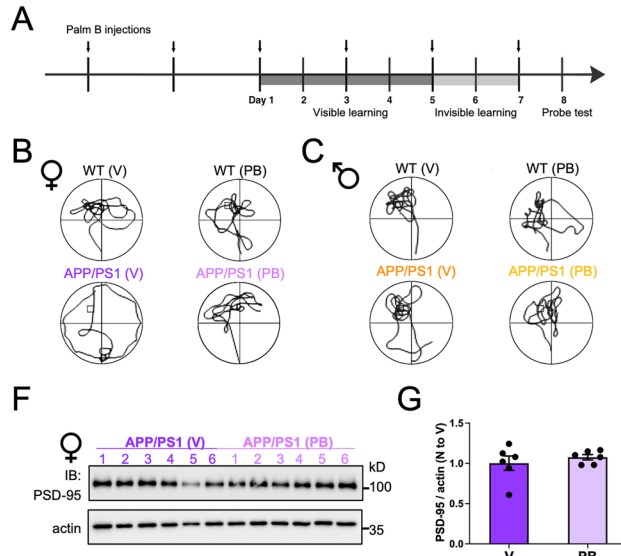

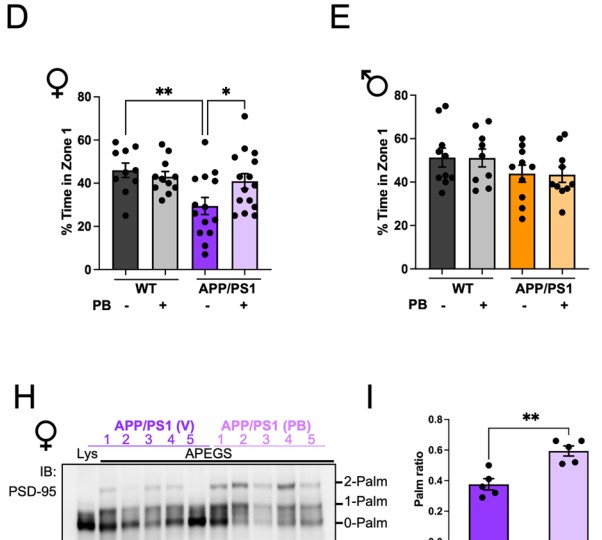

**Fig. 2 | Palm B rescues memory deficits and PSD-95 palmitoylation in 9–10-month-old female APP/PS1 mice. A** Time course of Palm B injections and Morris water maze behavioral testing. See also Supplementary Fig. 7. **B** Representative trajectories measured during the probe test for female mice treated with vehicle (V) or Palm B (PB). **C** Representative trajectories measured during the probe test for male mice. **D** % of time spent in zone 1 for female mice. $N = 10$ mice for WT (V), $N = 11$ for WT (PB), $N = 14$ for APP/PS1 (V), $N = 15$ for APP/PS1 (PB). * $p < 0.05$, ** $p < 0.01$, Two-way ANOVA (effect of genotype and treatment, $F (3,32) = 4.5$, $p = 0.009$) followed by Fisher LSD post-hoc test. **E** % of time spent in Zone 1 for male

mice. $N = 10$ mice for WT (V), $N = 9$ for WT (PB) $N = 10$ for APP/PS1 (V), $N = 10$ for APP/PS1 (PB). See also Supplementary Fig. 7. **F** Western blot of hippocampal lysates from female APP/PS1 mice injected with vehicle (V) or Palm B (PB). Immunoblotting with PSD-95 and actin. **G** Quantification of the Western blot shown in (**F**). $N = 6$ WT and APP/PS1 mice. Not significant, Mann–Whitney test. **H** Western blot of hippocampal samples from female APP/PS1 mice that underwent the APEGS assay. **I** Quantification of the Palm ratio in the Western blot shown in (**H**). $N = 5$ WT and APP/PS1 mice. ** $p < 0.01$, Mann–Whitney test. See also Supplementary Fig. 8. Error bars in all panels represent ± SEM.

thioesterase 1 (APT1)[22], we directly measured the palmitoylation of p62, a major APT1 target[51] in APP/PS1 mice. Palm B had no effects on total levels or palmitoylation of the p62 protein (Supplementary Fig. 10). Furthermore, we also quantified the effect of Palm B in WT animals and found that PSD-95 levels, as measured by Western blotting, were similar in vehicle and Palm B-injected animals of both sexes (Supplementary Fig. 11).

### Palmitoylation substrates are higher in male APP/PS1 mice
While several laboratories have reported sex differences in the APP/PS1 model[6,46–48], to our knowledge, sex differences in PSD-95 palmitoylation have not been reported previously and could contribute to the more severe cognitive impairments observed in female APP/PS1 mice compared to male mice[6–8]. Moreover, recent studies have described differences in metabolic pathways in women compared to men[52], suggesting that metabolites might differ between male and female mice as well. To gain further insight into the origins of the sex difference we observed, we performed metabolomic analysis of plasma from male and female APP/PS1 mice, treated with vehicle or Palm B (Fig. 3). Interestingly, we found that levels of palmitic acid, one of the main substrates for palmitoylation, were significantly higher in male mice: $3.0 \times 10^6 \pm 0.1 \times 10^6$ counts for male mice, $N = 10$; vs. $2.42 \times 10^6 \pm 0.09 \times 10^6$ counts for female mice, $N = 14$; $p < 0.01$ and that Palm B increased this metabolite, normalizing its levels for female APP/PS1 mice (Fig. 3B). Another significant palmitoylation substrate, stearic acid[23], had similar trends (Fig. 3C), suggesting that palmitoylation substrates are higher in male APP/PS1. In contrast, amino acids were similar across sexes and were not significantly different in Palm B-injected animals (Supplementary Fig. 12). To better understand how levels of fatty acids in plasma relate to their brain concentrations, we performed lipidomic analysis of free fatty acids (FFAs) in hippocampal tissue (Fig. 3D–F). We quantified the proportion of the 10 most abundant fatty acids (FFAs) detected using both approaches and found drastic differences in the proportion of saturated vs. unsaturated FFAs. As expected, saturated fatty acids represented a much higher proportion of the FFAs measured in hippocampal tissue than in plasma (Fig. 3D). Moreover, when we looked at palmitoylation substrates

specifically (C16 and C18, palmitic and stearic acid), we saw that the proportion of these FFAs in plasma was significantly higher in male mice: $18.3 \pm 0.6\%$ for female mice, $N = 14$; vs. $20.5 \pm 0.9\%$ for male mice, $N = 10$; $p < 0.05$ (Fig. 3E). In hippocampal tissue, the proportion of palmitoylation substrates was above 60% in vehicle treated mice and importantly we observed a significant reduction in Palm B treated animals in female APP/PS1 only: $71 \pm 5\%$ for vehicle treated female mice, $N = 6$; vs. $48 \pm 4\%$ for Palm B treated female mice, $N = 6$; $p < 0.001$ (Fig. 3F). This reduction in the levels of free palmitoylation substrates indicates that Palm B treatment leads to more palmitoylation as these fatty acids are attached to proteins. Furthermore, sex differences observed in FFAs levels in plasma and hippocampal tissue supports the sex-specific effects of Palm B and might contribute to sex differences in AD.

### Palm B does not affect amyloid plaques or astrocytes in APP/PS1 mice
Synaptic activity was shown to directly affect APP processing, with increased activity leading to more Aβ production and blockade of neuronal activity with TTX leading to a rescue of Aβ effects on AMPA-mediated synaptic transmission[53]. Because Palm B increased PSD-95[30,35] as well as surface GluA2[30] in cultured neurons, it is possible that Palm B injections would increase synaptic activity; consequently increasing Aβ production and AD pathology. To determine if Palm B injections have any effects on AD pathology, we analyzed amyloid plaques in the hippocampus using the commonly used stain thioflavin-S[48,54,55]. Moreover, astrocytes are known to be implicated in AD pathophysiology as well as Aβ clearance[56–59]. Increased expression of glial fibrillary acid protein (GFAP), expressed primarily in astrocytes, a major glial cell type in the CNS, is associated with AD progression[59], and GFAP deletion increased amyloid plaque build-up in AD mice[57]. We thus quantified astrocytes using IHC for GFAP in the same brain slices stained with thioflavin-S (Fig. 4). We found that the area covered by thioflavin-S-stained plaques was not affected by Palm B injections, both for female and male mice. Palm B also had no effect on the average plaque size; suggesting that in APP/PS1 mice, it does not lead to any effects on amyloid

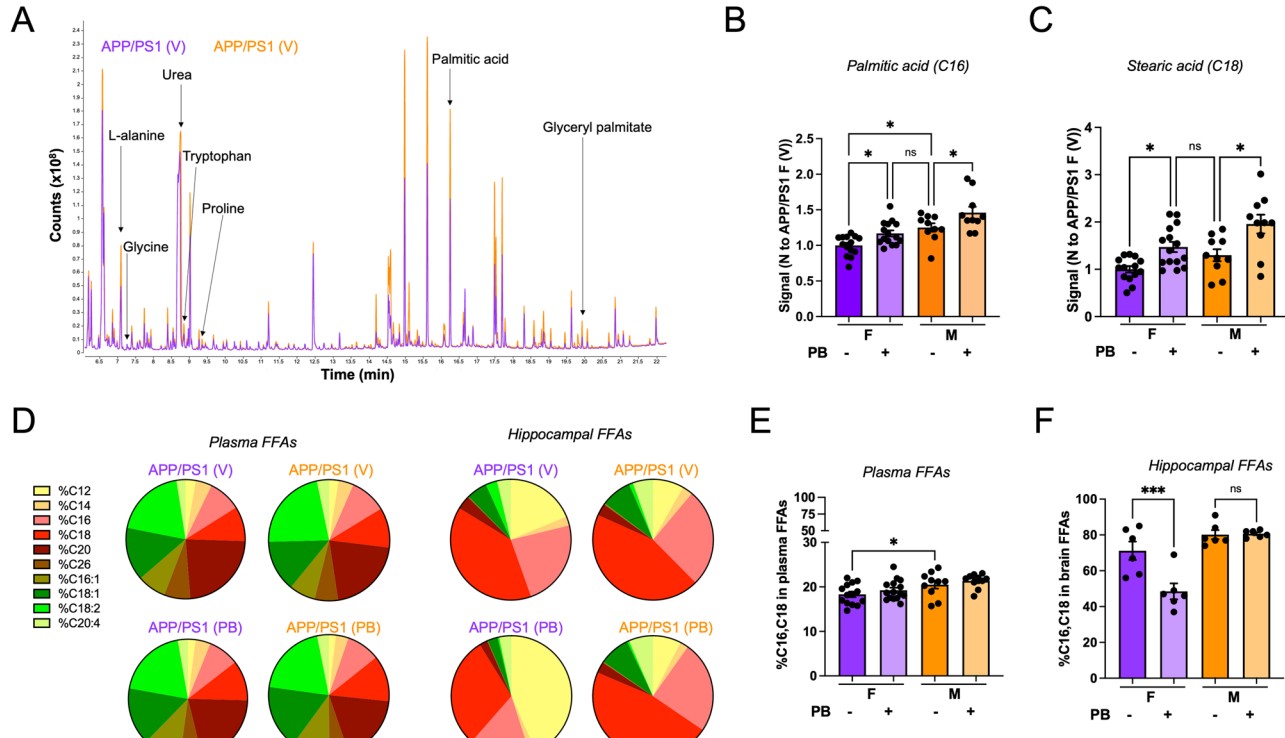

**Fig. 3 | Palmitoylation substrates are higher in male APP/PS1 mice.**
**A** Representative total chromatograms of plasma metabolites in female (purple) and male (orange) APP/PS1 mice treated with vehicle. See also Supplementary Fig. 10. **B** Quantification of palmitic acid levels in plasma samples from female ($N = 14$ for APP/PS1 (V), $N = 15$ for APP/PS1 (PB)) and male ($N = 10$ for APP/PS1 (V) and APP/PS1 (PB)) APP/PS1 mice treated with vehicle (V) or Palm B (PB). Two-way ANOVA (effect of sex and genotype, $F_{(3,31)} = 8.7$, $p = 0.0002$) followed by Fisher LSD post-hoc test. *$p < 0.05$, n.s. not significant. **C** Quantification of stearic acid levels in the same samples. Two-way ANOVA (effect of sex and genotype, $F$

$_{(3,31)} = 8$, $p = 0.0004$) followed by Fisher LSD post-hoc test. *$p < 0.05$. **D** Proportion of the 10 most abundant free-fatty acids (FFAs) in mouse plasma (left) and hippocampal tissue (right) for all groups tested. **E** Combined proportion of palmitic and stearic acid (relative to the 10 FFAs shown in **D**) in mouse plasma, same samples as in (**B**, **C**). Two-way ANOVA (effect of sex and genotype, $F_{(3,31)} = 3.4$, $p = 0.027$) followed by Fisher LSD post-hoc test. *$p < 0.05$. **F** Combined proportion of palmitic and stearic acid in hippocampal tissue. $N = 6$ mice for all groups. Two-way ANOVA (effect of sex and genotype, $F_{(3,15)} = 10$, $p < 0.0001$) followed by Fisher LSD post-hoc test. ***$p < 0.001$. Error bars in all panels represent ± SEM.

plaques. As previously reported[6,46–48], we observed sex differences in thioflavin-S-stained amyloid plaques. The area covered by plaques was significantly higher in female APP/PS1 treated with Palm B than for males: $0.45 \pm 0.05\%$ for female mice, $N = 16$; vs. $0.34 \pm 0.05$ for male mice, $N = 12$; $p < 0.05$ (Fig. 4C). We also observed that the average plaque size was higher in male mice treated with vehicle (or Palm B) than in female mice: $230 \pm 14\ \mu m^2$ for male mice treated with vehicle, $N = 12$; vs. $174 \pm 8\ \mu m^2$ for female mice treated with vehicle, $N = 15$; $p < 0.01$ (Fig. 4D). Female APP/PS1 mice had a smaller hippocampus, as measured by a decreased hippocampal area in vehicle treated mice: $2.0 \times 10^6 \pm 0.1 \times 10^6\ \mu m^2$ for female mice, $N = 11$; vs. $2.27 \times 10^6 \pm 0.08 \times 10^6\ \mu m^2$ for male mice, $N = 12$; $p < 0.05$ (Fig. 4E). Palm B injections had no effect on hippocampal area measurements in either male or female APP/PS1 mice (Fig. 4E). Similarly as Kraft et al.[57], in both vehicle and Palm B treated animals we saw GFAP signal gathering around amyloid plaques (Fig. 4A, B). In contrast, the GFAP signal was uniformly distributed in WT mice (Supplementary Fig. 13). Furthermore, GFAP immunoreactivity was similar in APP/PS1 mice of both sexes, and Palm B had no effect at all on GFAP signal (Fig. 4F). These results indicate that Palm B has no direct effects on amyloid plaques or astrocyte activation in 9–10 months old APP/PS1 mice.

### Palm B rescues impairments in synaptic transmission and dendritic spines in female APP/PS1 mice
Pharmacological blockade of PSD-95 depalmitoylation with Palm B reversed synaptic damage induced by Aβ in cultured hippocampal slices[35]. To test if Palm B has a similar protective effect on synaptic transmission in APP/PS1 mice, we recorded and analyzed the properties of AMPA receptor

(AMPAR)-mediated miniature excitatory synaptic currents (mEPSCs) of CA1 hippocampal pyramidal neurons in acute brain slices from female WT and APP/PS1 mice injected with vehicle or Palm B (Fig. 5A–F). These animals received the same injection protocol as shown in Fig. 2A, and were used for recordings within 24 h after the last injection. We analyzed mEPSCs frequency by quantifying the inter-event interval, as it can inform on the relative number of synapses mediating synaptic transmission[60]. We found that the mEPSC inter-event interval was significantly higher for APP/PS1 mice than for WT mice: $2.4 \times 10^3 \pm 0.5 \times 10^3$ ms for WT, $n = 68$ recorded cells; vs. $5 \times 10^3 \pm 1 \times 10^3$ ms for APP/PS1 mice, $n = 64$; $p < 0.01$ (Fig. 5C), indicating weakened spontaneous synaptic transmission in female APP/PS1 mice. This is consistent with previous studies in APP/PS1[61] and APP knock-in mice[62]. This deficit was fully rescued in Palm B-injected animals, which was evident by looking at the significant difference of mEPSC inter-event interval between APP/PS1 mice injected with vehicle or Palm B: $5 \times 10^3 \pm 1 \times 10^3$ ms in the vehicle group, $n = 64$; vs. $2.1 \times 10^3 \pm 0.3 \times 10^3$ ms in the Palm B group, $n = 41$; $p < 0.01$ (Fig. 5C). In addition, we analyzed the cumulative frequency distribution of inter-event interval from each recorded mEPSC event of each recorded neuron. The cumulative distribution curves also indicated a higher inter-event interval in APP/PS1 mice, which was restored by Palm B injections (Fig. 5F). In mice injected with vehicle, we found that mEPSC amplitude was higher for APP/PS1 mice than for WT mice: $-10.5 \pm 0.5$ pA for APP/PS1 mice, $n = 64$; vs. $-8.7 \pm 0.4$ pA for WT, $n = 68$; $p < 0.01$ (Fig. 5D). Interestingly, an increase in mEPSC amplitude was also observed in 5xFAD mice[63]. Larger mEPSC amplitude usually signifies a stronger postsynaptic response to a single vesicle release[64,65], suggesting that larger synapses are more abundant in the hippocampus of APP/

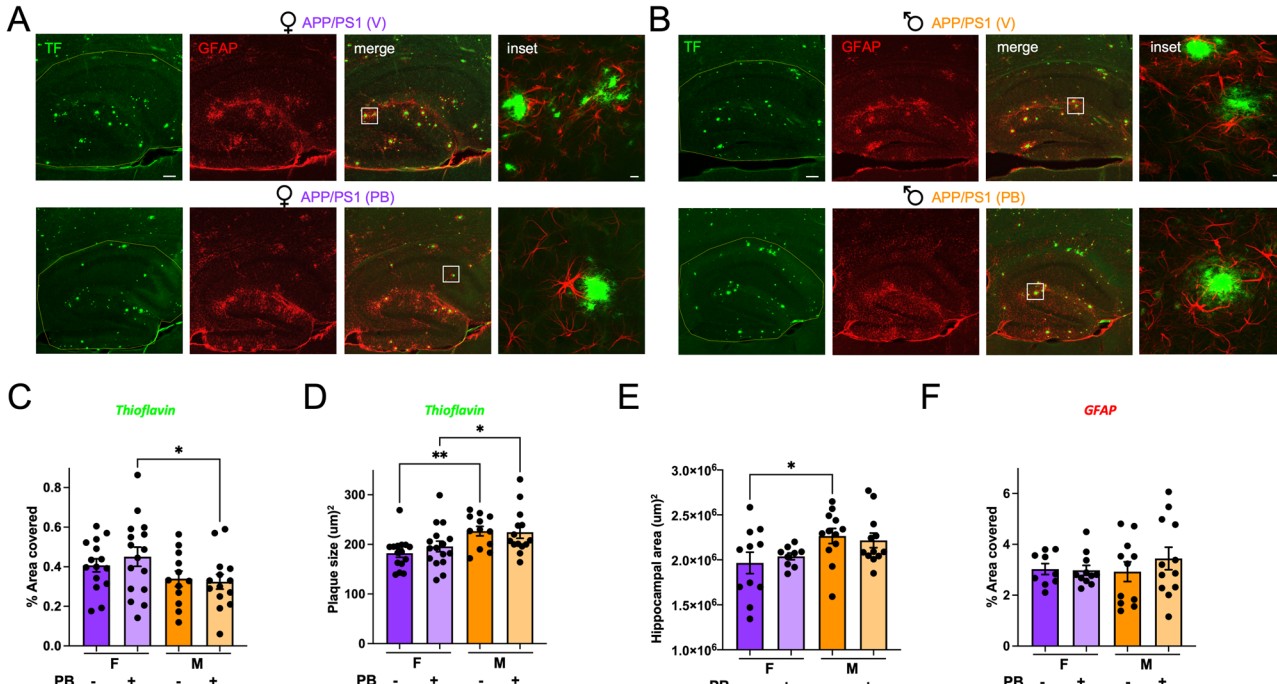

**Fig. 4 | Palm B does not affect amyloid plaques or astrocytes in APP/PS1 mice.**
**A** Representative example of APP/PS1 female mice hippocampus treated with vehicle (V) (top), or Palm B (PB) (below) stained with thioflavin-S (left, hippocampal area outlined in yellow) and GFAP (middle). Scale bar is 200 μm. Inset (right) shows high magnification image of the area outlined in white in the merged image, scale bar is 10 μm. **B** Representative example of APP/PS1 male mice hippocampus treated with vehicle (V) (top), or Palm B (PB) (below) stained with thioflavin-S (left) and GFAP (middle). Inset (right) shows high magnification image of the area outlined in white in the merged image. **C** Quantification of the % area covered by plaques in female ($N = 15$ for APP/PS1 (V) and $N = 16$ for APP/PS1 (PB)) and male ($N = 12$ for APP/PS1 (V) and $N = 14$ for APP/PS1 (PB)) APP/PS1 mice treated with vehicle (V) or Palm B (PB), 1 section per animal, Bregma = 1.56 mm. Two-way ANOVA (effect of sex and genotype, $F_{(3,38)} = 2.9$, $p = 0.049$) followed by

Fisher LSD post-hoc test. *$p < 0.05$. **D** Quantification of the average size of thioflavin-S-stained plaques in female and male APP/PS1 mice treated with vehicle (V) or Palm B (PB) in the same sections as in (**C**). Two-way ANOVA (effect of sex and genotype, $F_{(3,38)} = 7.6$, $p = 0.0004$) followed by Tukey post-hoc test. *$p < 0.05$, **$p < 0.01$. **E** Quantification of the hippocampal area in female ($N = 11$ for APP/PS1 (V) and $N = 9$ for APP/PS1 (PB)) and male ($N = 12$ for APP/PS1 (V) and $N = 12$ for APP/PS1 (PB)) APP/PS1 mice treated with vehicle (V) or Palm B (PB). Two-way ANOVA (effect of sex and genotype, $F_{(3,29)} = 3.2$, $p = 0.034$) followed by Fisher LSD post-hoc test. *$p < 0.05$. **F** Quantification of the % area covered by astrocytes in female ($N = 9$ for APP/PS1 (V) and $N = 11$ for APP/PS1 (PB)) and male ($N = 11$ for APP/PS1 (V) and $N = 12$ for APP/PS1 (PB)) APP/PS1 mice treated with vehicle (V) or Palm B (PB). Two-way ANOVA (effect of sex and genotype, $F_{(3,28)} = 0.42$, $p = 0.74$). Error bars in all panels represent ± SEM.

---

PS1 mice in comparison to WT mice, as would be expected if Aβ led to the removal of small synapses. Notably, we found that mEPSC amplitude was significantly lower in APP/PS1 mice injected with Palm B than in APP/PS1 mice injected with vehicle: $-9.5 ± 0.6$ pA for the Palm B group, $n = 41$; vs. $-10.5 ± 0.5$ pA for the vehicle group, $n = 64$ $p < 0.05$ (Fig. 5D), suggesting that Palm B normalized synapses in APP/PS1 mice. Importantly, Palm B injections had no effect on mEPSC amplitude or inter-event interval in WT mice (Fig. 5C–F), indicating that this drug does not affect synaptic transmission in normal neurons.

Changes in mEPSC frequency and amplitude are often related to changes in the structure of dendritic spines[66–68]. Therefore, we examined dendritic spine morphology and density by staining acute slices with the lipophilic dye DiI. This method resulted in sparsely labeled neurons throughout the brain slices. Consistently, with other experiments in this study, we focused on pyramidal neurons located in the CA1 region of the hippocampus (Fig. 5G) and analyzed dendritic spines in apical dendrites ~100–200 μm away from the cell body. Several studies reported reductions in dendritic spine density in APP/PS1 mice[40–43]. As expected, we also found a significant decrease of spine density in female APP/PS1 mice compared to WT mice, which was rescued in Palm B-injected female APP/PS1 mice (Supplementary Fig. 14A, B). To investigate if smaller dendritic spines are more vulnerable to AD pathology, we analyzed their size distribution in different conditions and saw that the proportion of large dendritic spines was significantly higher in APP/PS1 mice and that Palm B treatment normalized the distribution (Fig. 5I). Accordingly, we found that in female APP/PS1 mice, dendritic spines were significantly larger: $0.144 ± 0.004$ μm²

in WT mice, $N = 132$ dendrite sections; vs. $0.216 ± 0.008$ μm² in APP/PS1 mice, $N = 120$; $p < 0.0001$ (Fig. 5J). These results are consistent with the electrophysiological recordings described above, which showed that female APP/PS1 mice exhibited larger mEPSC amplitudes. Accordingly, our study found that dendritic spines with larger size and greater mEPSC amplitudes were more prevalent in APP/PS1 mice. Importantly, Palm B injections reversed this increased spine size in female APP/PS1 mice: $0.216 ± 0.008$ μm² for the vehicle group, $N = 120$; vs. $0.142 ± 0.006$ μm² for the Palm B group, $N = 118$; $p < 0.0001$ (Fig. 5G–J). To further demonstrate that small dendritic spines are specifically targeted in female APP/PS1 mice, we analyzed the density of two categories of dendritic spines, small and large (using the median size measured in WT female mice). Spine density of dendritic spines smaller than 0.13 μm² was significantly decreased in APP/PS1 mice as compared to WT mice: $0.86 ± 0.05$ spines/μm in WT mice, $N = 132$ dendrite sections; vs. $0.32 ± 0.04$ spines/μm in APP/PS1 mice, $N = 120$; $p < 0.0001$, Fig. 5K. This effect was completely rescued in animals injected with Palm B. No significant difference was found between WT and APP/PS1 mice when analyzing the density of large dendritic spines, and Palm B had no effect in those spines in APP/PS1 mice (Fig. 5L). Instead, Palm B reduced the density of large dendritic spines in WT mice, possibly due to unspecific effects of Palm B when PSD-95 palmitoylation is maximal (Fig. 5L). To investigate if sex differences in dendritic spines contribute to the sex differences seen in this study, we also used the same approach to characterize dendritic spines in male mice. Similarly, as in female mice, we found that spine density was decreased in male APP/PS1 mice, which went along with an increase in spine size (Supplementary Fig. 14C, D, F).

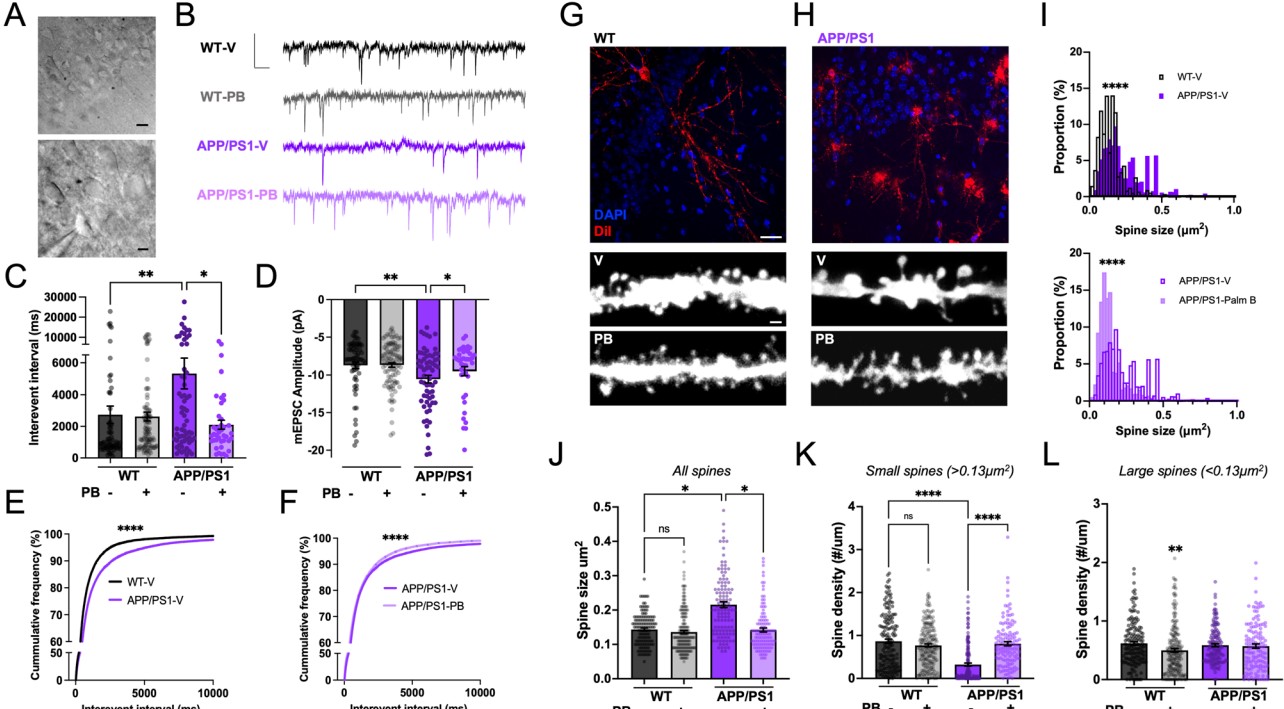

**Fig. 5 | Palm B rescues impairments in synaptic transmission and dendritic spines in female APP/PS1 mice. A** (Top) *Stratum pyramidale* of the hippocampal CA1 region in an acute slice of a 10-month-old WT female mouse. Scale bar is 20 μm. (Bottom) example of a CA1 pyramidal neuron with patch pipette forming whole-cell access. Scale bar is 5 μm. **B** Representative mEPSCs recording traces of hippocampal CA1 neurons from WT and APP/PS1 mice treated with vehicle (V) or Palm B (PB). Scale is 200 ms, 10 pA. **C** Quantification of mEPSC interevent interval from female WT and APP/PS1 mice treated with vehicle (V) or Palm B (PB). $N = 8$ mice, $n = 68$ recorded cells for WT (V), $N = 10$, $n = 96$ for WT (PB), $N = 10$, $n = 64$ for APP/PS1 (V), $N = 6$, $n = 41$ for APP/PS1 (PB); each data point is a single recording from an individual neuron. Two-way ANOVA (effect of genotype and treatment, $F(3,112) = 5.22$, $p = 0.002$) followed by Fisher LSD post-hoc test. $**p < 0.01$, $*p < 0.05$. **D** Quantification of mEPSC amplitudes in the same cells as in (**C**). Two-way ANOVA (effect of genotype and treatment, $F(3,112) = 4.047$, $p = 0.009$) followed by Fisher LSD post-hoc test. $**p < 0.01$, $*p < 0.05$. **E**, **F** Cumulative frequency (%) of mEPSC interevent interval from female WT treated with vehicle (V, $n = 23298$ events) and APP/PS1 mice treated with vehicle (V, $n = 21262$ events) or Palm B (PB,

$n = 22124$). $****p < 0.0001$, Kolmogorov–Smirnov test. $D = 0.09003$, $p < 0.0001$ in (**E**), $D = 0.10135$, $p < 0.0001$ in (**F**). (Top) representative images of DiI (red) and DAPI (blue) labeling in the CA1 of female WT (**G**) and APP/PS1 mice (**H**). Scale bar is 20 μm. (Bottom) representative high magnification images of dendrite and dendritic spines from female WT and APP/PS1 mice treated with vehicle (V) or Palm B (PB), scale bar is 1 μm. See also Supplementary Fig. 14. **I** Histograms showing different spine size distribution between WT ($n = 4244$ spines) and APP/PS1 ($n = 3084$ spines) female mice (upper); and between vehicle ($n = 3084$ spines) and Palm B injected APP/PS1 ($n = 2618$ spines) female mice. Kolmogorov–Smirnov test. $****p < 0.0001$. **J** Quantification of the average spine size. $N = 132$ for WT (V), $N = 199$ for WT (PB), $N = 120$ for APP/PS1 (V), $N = 118$ for APP/PS1 (PB); each data point is an individual dendrite section. Nested ANOVA, type II Wald $p$-value of genotype:treatment $= 0.047$; Tukey-adjusted pairwise post hoc comparisons, $*p < 0.05$. Quantification of spine density with small **K** and large **L** spines separated in the same dendrite sections as in (**J**). Two-way ANOVA (effect of genotype and treatment) followed by Fisher's LSD post-hoc test. $****$, $p < 0.0001$, $**p < 0.01$. Error bars in all panels represent ± SEM.

Therefore, in accordance with other studies suggesting that smaller spines are more vulnerable to AD pathology[13,35], we found that the density of small dendritic spines was drastically reduced in female APP/PS1 mice. Importantly, Palm B treatment rescued these smaller spines, restoring overall dendritic spine size toward normal levels.

## Discussion

Despite the recent approval of new anti-amyloid drugs for the treatment of AD, there is still uncertainty regarding their efficacy to improve cognition in patients[69]. Furthermore, cognitive reserve, which is related to higher levels of education and intellectual activity throughout a lifetime, was shown to be correlated with lower AD risk and slower disease progression[70,71]. Therefore, drugs promoting increased synaptic activity and stability have significant potential. Here, we show that female APP/PS1 animals have reduced PSD-95 palmitoylation as well as synaptic and cognitive deficits. Systemic injections of Palm B rescued PSD-95 palmitoylation as well as the synaptic and cognitive deficits observed in female APP/PS1 mice. In male APP/PS1 mice, we saw no reductions in palmitoylated PSD-95 and no behavioral deficits, suggesting that PSD-95 palmitoylation is correlated with hippocampal memory. Interestingly, a recent study found striking sex differences in the palmitoylation of synaptic proteins[72]. In female mice, gene ontology

revealed that the largest proportion of genes regulated by the DHHC7 palmitoylating enzyme were in involved in the "maintenance of synapse structure" while in male mice the same analysis found that genes involved in "metabolic processes" were dominating[72]. This suggests that in female mice, palmitoylation of synaptic proteins is important for maintaining dendritic spine integrity, while this is not the case in male mice.

Sex differences in AD have been extensively studied, both in humans[73] and in AD mouse models[6,46–48]. However, the mechanisms leading to sex differences are unclear. It is well known that Aβ effects on synapses require the NMDA receptor[74], and we demonstrated that increased PSD-95 (induced by Palm B treatment) protects synapses by blocking NMDA receptor metabotropic signaling[35]. Recent studies described sex differences in synaptic plasticity and the sensitivity to NMDA receptor antagonists[75–77]. Therefore, it is possible that these mechanisms are involved in mediating sex differences in AD. Specifically, distinctive NMDAR signaling might reduce palmitoylated PSD-95 in female APP/PS1 mice without affecting male APP/PS1 mice. Interestingly, sex differences in metabolic pathways in patients with AD and healthy controls were previously reported[52,78]. We detected significantly higher levels of palmitoylation substrates in the plasma of male APP/PS1 mice, which could promote palmitoylation of synaptic proteins, including PSD-95. Therefore, sex differences in protein palmitoylation[72],

NMDAR signaling[75–77] and lipid metabolism[52,78] are likely contributing to the sex differences seen in our study and the ones observed in most AD mouse models[6,46–48].

An important limitation of our study is that Palm B is not a specific inhibitor of ABHD17 enzymes, and has significant activity towards APT1 and APT2 as well as lower affinity for a few other protein depalmitoylases[34]. Therefore, it is likely that the palmitoylation of other proteins is also increased in Palm B-injected mice. However, we found that palmitoylation of p62, a major APT1 target was not affected in Palm B-injected animals. Also, Palm B had no effects on the behavior of WT male and female mice in the Morris water maze test (Fig. 2), indicating that possible unspecific effects had no profound consequences. Another important factor to consider is that palmitoylation of APP was shown to increase Aβ production[79]. Moreover, transgenic AD mice with unpalmitoylable cysteine residues on beta-secretase (BACE1) also had reduced amyloid burden and less cognitive deficits then the same AD mice without mutated BACE1[80]. Additionally, a recent study found increased protein palmitoylation in 3xTg AD model mice and that using the palmitoylation inhibitor 2-bromopalmitate rescued deficits in these mice[81]. These studies suggest that palmitoylation is involved in AD pathology and that increasing palmitoylation might not be beneficial for all proteins. To directly test if Palm B had any effects on AD pathology, we quantified amyloid plaques in APP/PS1 mice treated with vehicle or Palm B (Fig. 4) and found no effect of the drug on AD pathology; suggesting that Palm B does not affect APP or BACE1 palmitoylation. Moreover, Palm B did not have any effects on mEPSCs in female WT mice (Fig. 5C, D), consistent with our previous study where we saw no effects of Palm B on synaptic transmission in PSD-95KO mice[35]. PPT1 is another depalmitoylating enzyme known to affect synaptic proteins, including the GluA1 subunit of the AMPA receptor[82], however, Palm B has only low affinity for this enzyme[83], we thus expect little off-target effects linked to PPT1. Another limitation of our study is related to the use of the commercially available PF11 antibody. To our knowledge, all other published immunofluorescence images in neurons or brain slices were obtained with the PF11 antibody made by the Fukata laboratory[24,31,84]. We acknowledge that there is less overlap between PSD-95 and PF11 signals in our images, which indicates that some of the PF11 signal comes from unspecific staining and that the commercially available PF11 should be used with caution. However, as with the biochemical experiments shown in Figs. 1 and 2, we consistently measured a reduction in PF11 signal intensity in the hippocampus of female APP/PS1 mice (Supplementary Fig. 4) and a rescue of the PF11 signal in Palm B-injected mice (Supplementary Fig. 9).

Importantly, we observed that Palm B had direct effects on synaptic structure and function by recording mEPSCs and imaging dendritic spines in acute brain slices. Consistent with previous literature[61–63], hippocampal CA1 neurons in APP/PS1 female mice had larger mEPSC amplitudes and longer inter-event intervals as compared to WT mice. Both mEPSC amplitudes and inter-event intervals were rescued in Palm B-injected APP/PS1 female mice. Morphological analysis of dendritic spines revealed that dendritic spines were larger in female APP/PS1 mice, consistent with the observed increased mEPSC amplitude. Interestingly, changes in spine size have also been observed during normal aging in the CA1 region of the hippocampus. Xu et al. reported that thin, small spines were selectively lost in aged mice, while large, mushroom-type spines were more stable[85]. Comparable findings have been documented in the prefrontal cortex of monkeys[86]. This supports the notion that the loss of dendritic spines in female APP/PS1 mice is primarily from smaller spines, as we have shown that the density of small dendritic spines was specifically reduced without any effects in large dendritic spines as well as longer inter-event intervals of mEPSCs and a concomitant increase in average spine size, observed as larger mEPSC amplitudes in our electrophysiological assessment of hippocampal CA1 neurons. Interestingly, the density of small dendritic spines was also reduced in male APP/PS1 mice, suggesting that small dendritic spines are vulnerable in both male and female in this AD model.

AD is a synaptic disease first[11–13] and disproportionally affects women. In this study, we developed a pharmacological approach that results in increased palmitoylated PSD-95, specifically stabilizing dendritic spines vulnerable to AD-related degeneration. Notably, this increase in palmitoylated PSD-95 reversed memory and synaptic deficits found specifically in female AD model mice. Hence, this new therapeutic strategy may be beneficial for both treating and preventing AD in the population that suffers the consequences of this disease the most.

## Methods

### Mice
Male and female APP/PS1 AD model mice (MMRRC Strain #034829-JAX, background strain C57BL/6J) and their WT littermates aged between 9 and 10 months were used for all experiments. APP/PS1 mice are double transgenic AD model mice expressing a chimeric mouse/human amyloid precursor protein with the Swedish mutation (APP695swe) and a mutant human presenilin 1 (PSEN1dE9), causing early disease onset[87]. APP/PS1 mice were obtained from Jackson Laboratories and bred with C57BL/6J mice (also obtained from Jackson Laboratories, #000664-JAX) in order to maintain a live colony. Mice were kept in the UCSD School of Medicine facility on a 12-h light-dark cycle, given ad libitum access to food and water, and their genotypes were confirmed using PCR genotyping. For all experiments described below, mice were euthanized by isoflurane inhalation and decapitated after deep anesthesia was confirmed by unresponsiveness to paw pinch. All animal procedures were approved by UCSD' IACUC, and we confirm that we have complied with all relevant ethical regulations for animal use.

### Palmostatin B injections
APP/PS1 and WT mice were randomized to receive either systemic palmostatin B (Palm B, 178501, Sigma, 19.92 mM in DMSO then 1:9 dilution with $H_2O$; 5 mg/kg i.p.) or vehicle injections (10% DMSO in $H_2O$) every other day for total six injections. For behavioral tests, the experiments started on the day after the second injection and were performed on the days between injections. Brains were harvested within 24 h after the last injection. Brains were cut in half along the midline, and hemi-brains were either flash-frozen in a mixture of dry ice and isopropanol for Western blotting or drop-fixed in 4% PFA for immunohistochemistry. For electrophysiology and DiI labeling, acute slices were made from fresh brains, see the "acute brain slice preparation" section below.

### Western blotting
Hemibrains were dissected into four regions: frontal cortex, mid-brain, hippocampus, and posterior cortex. Each brain section was homogenized in buffer B (4% SDS, 0.5 M EDTA, 8.9 M Urea in PBS), supplemented with Protease Inhibitor Cocktail (P8349, Sigma). Samples were then centrifuged for 10 min, supernatant was taken, and protein concentration measured using the Pierce BCA Assay (#23335, ThermoFisher Scientific). Protein concentration was normalized to 0.75 mg/mL in 500 μL of buffer A (4% SDS, 0.5 M EDTA, in PBS), supplemented with Protease Inhibitor Cocktail (Sigma) and PMSF (0.5 mM, P7626, Sigma). Lysates were prepared for all samples with 2X Laemmli Sample buffer (Biorad). To quantify PSD-95 palmitoylation, we used the Acyl-PEGyl exchange gel shift (APEGS) assay exactly as described in this protocol[45]. Briefly, samples were incubated with 25 mM TCEP (C4706, Sigma) for 1 h at room temperature. To block free cysteine residues, NEM (E3876, Sigma) was added (final 50 mM), and samples were incubated with gentle rocking for 3 h. Excess TCEP and NEM were removed by a chloroform-methanol precipitation (as described by Kanadome et al.[45]). The resulting pellet was resuspended and treated with 1 M hydroxylamine ($NH_2OH$, 159417, Sigma) for 1 h at 37 °C. A second chloroform-methanol precipitation was performed, and protein concentration was measured using the BCA assay. Samples were normalized to a concentration of 0.75 mg/mL in 100 μL of Buffer A and treated with 30 μL of 600 mg/mL maleimide-conjugated poly-ethylene glycol (10k-mPEG, SUNBRIGHT ME-100MA, NOF corporation), final 15 mM for 1 h, to label palmitoylated cysteine residues. A final chloroform-methanol precipitation was performed to remove excess mPEG, and the pellet was suspended in

Laemmli Sample Buffer. Lysate samples (~10 µg of protein/sample) were separated on a 10% acrylamide: bisacrylamide gel, and samples resulting from the APEGS assay (~20 µg of protein/sample) on a 7.5% acrylamide: bisacrylamide gel. All samples were transferred into PVDF membranes and blocked in 5% milk in TBST for 1 h at room temperature. The primary antibodies; PanPalm (anti-pan palmitoylated cysteine polyclonal antibody (CBL-PTM-pal, Creative Biolabs)) at a 1:1000 dilution, PSD-95 (#MA1-045, Thermo-Fisher) at a 1:1000 dilution and β-actin (#3700, cell signaling) at a 1:2000 dilution) were incubated with gentle rocking at 4 °C overnight in 5% milk in TBST. The secondary antibodies (anti-mouse IgG, #7076, Cell Signaling Technology (CST); anti-rabbit IgG, #7074, CST) were incubated with gentle rocking at room temperature for 1 h (1:2000 dilution in 5% milk in TBST). Clarity ECL substrates (Biorad) were used for chemiluminescence detection. Protein bands were analyzed with the "gel analyzer" tool in FIJI (ImageJ). To measure PSD-95 palmitoylation, we looked at three bands: 150 kD, corresponding to two palmitoylated cysteine (2-Palm), 120 kD for one cysteine (1-Palm), and 95 kDa for no palmitoylation (0-Palm). The palm ratio was calculated by adding the 2-Palm and 1-Palm bands and then divided by the sum of all three bands. Similarly, the 1-Palm and the 2-Palm ratios were calculated by dividing the intensity of those bands by the sum of all three bands. For lysate samples, the PSD-95 95 kD band was normalized by the β-actin signal at 42 kD. For the p62 data shown in Supplementary Fig. 8, primary antibodies; p62 (5114BC, CST) at a 1:500 dilution and tubulin (6074, Sigma) a 1:10,000 dilution were incubated overnight at 4 °C in 1% bovine serum albumin (BSA) in TBST. All uncropped and unedited Western blotting images are provided in the Supplementary Information document.

### Immunohistochemistry (IHC) and thioflavin-S staining

After 2 days in 4% PFA, hemibrains from 9 to 10-month-old WT and APP/PS1 mice were transferred to PBS and sliced on a Leica vibratome (VT 1000S) to obtain 50 µm sagittal slices. Slices were stored in Cryo solution (30% ethylene glycol, 30% glycerol, 40% PBS) at −20 °C until ready to use. For IHC, slices (one per animal) were rinsed in PBS and treated with 50% formic acid (Sigma) in PBS for 5 min for antigen retrieval. After 3 PBS washes, slices were blocked with PermBlock (0.2% Triton X-100, 10 mg/mL BSA, 2% goat serum in PBS) for 1 h at room temperature. Primary antibodies diluted in PermBlock at a concentration of 1/300 for human anti-PF11 (#AG-27B-0021-C100, AdipoGen), 1/150 for rabbit anti-PSD-95 (#51-6900, Invitrogen) and 1/1000 for chicken anti-MAP2 (#CPCA-MAP2, EnCor) were incubated 4 °C overnight with gentle rocking. In separate experiments (Fig. 4 and Supplementary Fig. 3), we used 1/1000 for rabbit anti-GFAP (#MAB360, Sigma Aldrich) or 1/500 for rabbit anti-GluA3 (#AGC-010, Alomone labs). After three washes, secondary antibodies at a 1/1000 dilution in PermBlock were applied with gentle rocking for 1 h at room temperature. For PSD-95 and GFAP, an anti-rabbit antibody labeled with Alexa Fluor 647 was used (#4410, Cell Signaling). For PF11, an anti-human antibody labeled with Dylight 550 was used (#ab98820, Abcam). For MAP2, we used an anti-chicken antibody labeled with Alexa Fluor 488 (#A11039, Invitrogen) and an anti-rabbit antibody labeled with Alexa Fluor 488 (#A11034, Invitrogen) to detect GluA3. After three washes, slices were placed onto Superfrost Plus Slides (Fisher) and left to dry. Slides were then dehydrated (2 min 70% ethanol, 2 min 90% ethanol) and treated with 100% xylenes (Fisher) for 5 min, mounted (Eukitt quick-hardening mounting medium #03989 (Sigma Aldrich)) and coverslipped (no. 1 cover glass, #48393 106, VWR). For thioflavin-S staining, after xylenes treatment, slides were rehydrated (2 min 100% ethanol, 2 min 90% ethanol, 2 min 70% ethanol) and stained with 0.025% thioflavin-S in 50% ethanol for 10 min. Following staining, slices were washed with 70% EtOH for 5 min three times, dehydrated, treated with xylenes and mounted as above. Image acquisition was done in the CA1 region of the hippocampus with a Leica Stellaris 5 confocal microscope and a 60× oil objective for MAP2, PF11, PSD95, or GluA3 IHC. For the thioflavin-S and GFAP experiments shown in Fig. 4, we used a 5× air objective in order to visualize the whole hippocampus. To determine whether the PF11 signal observed on dendrites was

an artifact of the slice thickness used for IHC, we performed an experiment using 8 µm thin sections fixed in acetone[84] and observed only puncta (Supplementary Fig. 4D). Importantly, PF11 signal was significantly reduced in female APP/PS1 in these thin sections (Supplementary Fig. 4E). For these experiments shown in Supplementary Fig. 4D, brains were embedded in frozen OCT compound (23-730-625, Scigen), 8 µm sections were cut with a cryostat (NX50, Thermo Scientific) and allowed to dry on microscope slides completely at room temperature (RT). Then, sections were fixed in ice-cold acetone for 20 min and stored at −80 °C. For IHC, sections were washed in 25 mM PBS for 5 min, blocked in 10% goat serum for 30 min, incubated in primary antibodies (PSD-95 and PF11 as above) for 2 h at RT, in secondary antibodies for 1 h at RT and mounted in Prolong Gold Diamond mounting media (P36934, Thermo Fisher). Imaging was carried out with the Leica Stellaris 5 confocal as described above.

To quantify PSD-95 and PF11 in the CA1, we used ImageJ and measured MAP2, PF11 and PSD-95 fluorescence intensity in the same five regions of interest (ROI) in the dendritic layer (avoiding cell bodies and blood vessels). After subtracting the values obtained with a control slice (processed the same way but without primary antibodies), PF11 and PSD-95 values were normalized by the corresponding MAP2 ROI, then all five ROIs averaged to yield one value per slice. We note that slices with fluorescence intensities comparable to the negative control were excluded from analysis. For thioflavin-S and GFAP analysis, a mask outlining the hippocampus was traced for each brain slice, measuring the hippocampal area. The "particle analysis" function with optimized threshold (adjusted for each picture) and size exclusion values for thioflavin-S (20–1000 µm) or GFAP (10–1000 µm), respectively, was then used to quantify the % area covered by plaques or astrocytes as well as the average size of plaques. For the cryostat slices IHC shown in Supplementary Fig. 4D, E, we used a similar analysis method of drawing ROIs (square analysis) and also the "particle analysis" method using the PSD-95 channel to detect synaptic puncta (size exclusion values 0.05–2 µm, circularity 0.9–1) and quantify their intensity in both the PSD-95 and the PF11 channel.

### Morris water maze test

Morris water maze (MWM)[8,49,50,88] was used to test the effect of Palm B injections on spatial memory (Fig. 2A–E and Supplementary Fig. 7). Mice were tested in a 6-foot diameter tank filled with water mixed with white non-toxic paint to make the water opaque. Mice were given four 90 s trials per day for 7 consecutive days with an escape platform placed in zone 1; for days 1–3 the platform was visible with a flag, the flag was removed for days 4–5 and the platform was made invisible for days 6–7 (see Fig. 2A and Supplementary Fig. 7). The platform was removed from the tank for the probe test, consisting of one 40 s trial, done at day 8. Testing involved placing each mouse in the tank at water level, facing the pool wall, and at one of two start positions equidistant from the platform. Video tracking (ANY-maze) was initiated once the mouse was released and terminated automatically when the animal remained on the platform for more than 3 s. Mice were allowed to remain on the platform for a total of 20 s during the inter-trial interval. Animals that did not learn the location of the platform (never found the platform or if no reductions in escape latency were observed during training) were excluded from further analysis.

### Metabolite and lipidomic analysis

A novel vapor-phase extraction method was used to extract metabolites from mouse plasma (collected from APP/PS1 mice that underwent the Morris water maze test). The procedure began with 10 µL of flash-frozen, lyophilized plasma, homogenized in a frozen vial containing a monophasic solvent system composed of 1-butanol:acetonitrile in a 3:1 ratio. Samples were incubated at 37 °C for 30 min to enhance metabolite solubilization and recovery. Following incubation, samples were rapidly frozen in liquid nitrogen and subsequently lyophilized overnight to concentrate volatiles. The extracted metabolites were derivatized using a freshly prepared solution containing anhydrous acetonitrile, pyridine, and MSTFA (2:1:1). A two-stage heating process was applied post-derivatization to enhance the

stability and recovery of analytes before analysis. These prepared samples were subjected to gas chromatography-mass spectrometry (GC-MS) for comprehensive metabolite profiling, enabling sensitive and reliable detection of metabolites in complex plasma matrices.

Metabolite analysis was conducted using an Agilent 6890 series gas chromatograph coupled with an Agilent 5973 mass selective detector. The GC-MS parameters included the following: a DB-35 MS column (Agilent; 30 m × 250 μm × 0.25 μm film), mass source temperature of 230 °C, interface temperature of 250 °C, electron energy of 70 eV, and mass temperature of 150 °C. The method employed splitless injection of samples, with oven temperature program starting at 45 °C and held for 2.25 min. Subsequently, the temperature was ramped to 300 °C at 20 °C min$^{-1}$ and maintained at 300 °C for 5 min. A solvent delay of 4.5 min was included to protect the electron impact filament from damage by residual ethyl acetate in the samples.

Identification of metabolites was conducted by comparing observed mass spectra to authentic standards, as well as reference databases such as the Wiley, National Institute of Standards and Technology (NIST), and Adams libraries. This approach allowed accurate identification of both known and novel analytes. Metabolite peak areas were quantified using Agilent MassHunter quantitative analysis software, which was employed for peak integrations. MassHunter quantitative analysis method development was performed using the NIST library. Normalization of the data was performed using tetracosanoic acid as the internal standard, as detailed in the accompanying dataset (Supplemental Data 2). This normalization accounted for variability in sample preparation and instrumental performance.

FFA quantification in mouse hippocampal tissue (0.5 mg/ml of total protein as determined with the BCA assay) was performed at the UC San Diego Lipidomics Core[89] in 6 different animals of each condition. Results of this analysis are included in Supplemental Data 2. Ten most abundant FFAs present in both datasets were further analyzed in Fig. 3.

### Acute brain slice preparation
Adult mice (9–10 months old) from each group were anesthetized by isoflurane inhalation. After decapitation (note that no cardiac perfusion was used before decapitation), brains were gently extracted from the skull within 1 min and placed into pre-cooled and oxygenated (with 5% $CO_2$/95% $O_2$) NMDG-HEPES aCSF for 60 s. The NMDG-HEPES aCSF contains (in mM): 92 NMDG, 2.5 KCl, 1.25 NaH₂PO4, 30 NaHCO₃, 20 HEPES, 13 glucose, 2 thiourea, 5 Na-ascorbate, 3 Na-pyruvate, 12 N-acetyl-L-cysteine, 0.5 CaCl₂ and 10 MgSO₄ (pH 7.3–7.4, osmolarity 300–310 mOsm)[90]. The brain was cut at the midline and the two hemispheres were glued onto the slicing stage of the vibratome, the slicing chamber was filled with pre-cooled NMDG-HEPES aCSF and oxygenation was continuously provided throughout the procedure. 250–400 μm thick horizontal or sagittal brain slices were sectioned on the vibratome and carefully transferred into oxygenated NMDG-HEPES aCSF pre-warmed to 32 °C. The optimal Na+ spike-in method was used by adding 2 M NaCl stock solution for 5 times during the 10–30 min of slice recovery which gradually brought the NaCl concentration to 4 mM in the NMDG-HEPES aCSF[90]. After 35 min recovery in NMDG-HEPES aCSF, slices were transferred into oxygenated HEPES holding aCSF at room temperature and allowed to recover for an additional 1 h prior to initiating patch clamp recording experiments. The HEPES holding aCSF contained (in mM): 92 NaCl, 2.5 KCl, 1.25 NaH₂PO₄, 30 NaHCO₃, 20 HEPES, 13 glucose, 2 thiourea, 5 Na-ascorbate, 3 Na-pyruvate, 12 N-acetyl-L-cysteine, 2 CaCl₂, and 2 MgSO₄ (pH 7.3–7.4, osmolarity 300–310 mOsm).

### Electrophysiology (whole-cell patch clamp)
Slices were transferred unto the recording chamber with a continuous flow of recording aCSF containing 124 mM NaCl, 2.5 mM KCl, 24 mM NaHCO₃, 1.25 mM NaH₂PO4, 10 mM glucose, 2 mM CaCl₂, 2 mM MgSO₄, 1 μM tetrodotoxin and 10 μM gabazine (pH 7.4). The recording aCSF was perfused at a rate of 1.5–2.0 ml/min and oxygenated with 5% $CO_2$/

95% $O_2$ at 28–32 °C. Borosilicate glass pipettes (outer diameter: 1.5 mm, Warner, Hamden, CT) were pulled from a micropipette puller (Model P-97, Sutter). The recording electrodes had a resistance of 2–5 MΩ when filled with an internal solution containing (in mM): 115 cesium methanesulfonate (Sigma), 20 CsCl, 10 HEPES, 2.5 MgCl₂, 4 Na₂ATP, 0.4 Na₃GTP, 10 sodium phosphocreatine (Sigma), and 0.6 EGTA (Amresco) (pH 7.25, osmolarity 290–300 mOsm, all chemicals from Sigma unless otherwise specified). A MultiClamp 700B amplifier, an Axon Digidata 1550B, MultiClamp 700B software and Clampex 11 software (Molecular Devices, San Jose, CA, USA) were used for data acquisition, digitized at 2–10 kHz, and filtered at 2 kHz. CA1 pyramidal neurons were identified under differential interference contrast microscopy. The liquid junction potential was compensated for prior to forming the cell-attached mode for all recordings. A minimum of 2 GΩ seal resistance was required before rupturing the membrane for whole-cell configuration. Neurons were allowed a 3–5 min resting period after whole-cell formation. Miniature excitatory postsynaptic currents (mEPSCs) were recorded for 3-5 min under voltage clamp at a holding potential of −60 mV. The membrane capacitance (Cm), membrane resistance (Rm), and series or access resistance (Ra) were tested by "membrane test" in Clampex 11 before and after the mEPSCs recording for each cell. The recordings with Ra larger than 30 MΩ or with variations larger than 20% were excluded from data analysis. The peak amplitude, frequency, and inter-event interval time of mEPSCs were analyzed using "event detection-templated search" in blind software (Molecular Devices, San Jose, CA, USA). Template was created by extracting and averaging segments of data that are manually identified as corresponding to an mEPSC event, and match threshold 4–5 was used for event detection. Events with absolute value of peak amplitude smaller than 4 pA were excluded from quantification. The average values of mEPSC amplitude, frequency, and inter-event interval time from all the detected events of each recorded cell were used for mean value comparisons; and the individual values of mEPSC amplitude and inter-event interval time from all the detected events of each recorded cell were used for the cumulative frequency comparisons. Animals used for electrophysiology experiments were not used for behavioral testing or IHC experiments. Extra acute slices that were not used for electrophysiology were stained with DiI (see below).

### DiI labeling of dendrites
Acute brain slices were prepared as described above and fixed in 4% PFA in PBS for 60 min at room temperature with gentle agitation, then washed three times in PBS for 10 min. Slices were incubated with DiI solution (0.1% DiI in DMSO, diluted in PBS with 0.03% NaN₃ (1/500)) for 3 days with gentle agitation at room temperature in the dark. After three 30 min washes in PBS, slices were incubated with DAPI (5 μg/ml in PBS) for 30 min, washed one last time in PBS for 30 min, then mounted and coverslipped in ProLong Gold™ glass antifade mounting media (ThermoFisher). Slices were imaged within 24-48 h.

Image acquisition was conducted using a Leica STED Sp8 or Stellaris 5 confocal microscope. DAPI signal was used to identify the hippocampus and the CA1 region, low magnification images were first obtained with a 10× objective for each slice (see Fig. 5G, H for examples). High magnification images were then acquired using a 60× oil immersion objective, 0.3 μm step interval and a digital zoom factor of 6 from dendritic segments approximately 100–200 μm away from the neuronal soma in the *Stratum radiatum* layer. ImageJ was used to obtain Z-projection images and analyze spine density and spine size. For each image, the length of each straight dendritic segment was measured, and all dendritic spines were manually traced using an oval ROI (see Supplementary Fig. 14 for example analysis). Spine density was calculated by dividing the number of spines by the length of the dendritic segment, and spine size was estimated by the size of the oval ROIs and averaged for each dendritic segment. For the histograms of the observed spine sizes in female mice shown in Fig. 5I, all individual spines were analyzed. To determine if small dendritic spines were disproportionally affected in APP/PS1 mice, we used the median spine size in WT female mice (0.13 μm²) and counted the density of spines smaller and larger than

$0.13\ \mu m^2$ for each dendritic segment analyzed (Fig. 5K, L and Supplementary Fig. 14E, F).

## Statistics and reproducibility

Data reproducibility was ensured by conducting experiments in several number of animals. The Western blotting and immunohistochemistry experiments were also conducted several times and obtained similar results. A representative experiment is shown for all data presented. To determine sample size for behavioral experiments, we consulted with Robert Rissman, an expert in using AD mouse models, and decided on an $N = 10$–15. Immunohistochemistry, thioflavin staining and metabolomic data come from animals used in behavioral experiments. For biochemical experiments (Western blotting), electrophysiology and spine analysis shown in Fig. 5, we determined the required sample size from similar published studies. Mice were randomly selected for the treatment (Palm B injections) or the control (vehicle) groups. All data were analyzed blind to the experimental condition.

Statistical analysis was performed using Prism 10 (GraphPad Software). Means of two groups were compared using an unpaired, two-tailed t-test or the Mann–Whitney test for data with different standard distributions. One or two-way-ANOVA was used to compare multiple groups with one or two independent variables. Detailed information about the ANOVA statistics and the specific post-hoc test used can be found in the figure legends. Data are shown as mean ± SEM. For all analyses, $p \leq 0.05$ was considered significant. Data was graphed using GraphPad Prism 10.

For the dendritic spine analysis shown in Fig. 5 and Supplementary Fig. 14, we used a nested ANOVA to account for within-animal and within-dendrite variability. We used a linear mixed-effects model, with genotype and treatment included as fixed effects and a random intercept for dendrite segment nested within each mouse to account for within-animal and within-dendrite variability. Model residuals were assessed for normality using the D'Agostino–Pearson omnibus test. If residuals deviated significantly from normality, a Box–Cox transformation was applied to identify the most appropriate data transformation prior to refitting the model. Fixed effects were evaluated using type II Wald chi-square tests to examine main effects and interactions. When significant effects were detected, post hoc comparisons were conducted using estimated marginal means with Tukey-adjusted pairwise contrasts to identify specific group differences. Note that for the spine density data separated into small and large dendritic spines, this nested ANOVA method could not be used, due to the presence of 0 in the data; a two-way ANOVA was then used, as specified in the Figure legends.

## Reporting summary

Further information on research design is available in the Nature Portfolio Reporting Summary linked to this article.

## Data availability

All data generated or analyzed during this study are included in this published article (and its Supplementary Information and Supplementary Data files (all numerical Source data for graphs shown in the main Figures can be found in Supplementary Data 1–3). All other data are available from the corresponding author on reasonable request. Material requests should be sent to the corresponding author.

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

## Acknowledgements

We would like to thank Jazmin Florio and Michael Mante for behavioral testing training and help with tissue harvesting. We thank the late Dr. Eric Smeltz for giving Ahmed Khalil access to his GC/MS instrument. We also thank Drs. Roberto Malinow, Helmut Kessels, Christina Sigurdson and Alexandra Newton for their helpful comments on the manuscript. This work was funded by the National Institute on Aging, grant AG067049 to K.D. We also acknowledge grant P30NS047101 from NINDS, which funds the UCSD School of Medicine Microscopy core.

## Author contributions

Y.D. and K.P. performed the main experiments. A.Q.P. and M.D. contributed to electrophysiology data collection and analysis. A.L. did the thioflavin-S and GFAP experiments and contributed to writing the manuscript. C.M., M.D., M.G., and M.S. helped with biochemical experiments. A.K. did all the metabolomics experimentation and analysis. R.R. provided mice and access to Morris water maze equipment. M.M., I.B., and H.K. helped with data analysis and animal work. K.D. and Y.D. designed the research and wrote the manuscript. K.D. also performed behavioral testing, acquired funding and supervised all research. All authors contributed to the finalization of the manuscript.

## Competing interests

The authors declare no competing interests.

## Additional information

**Supplementary information** The online version contains Supplementary material available at https://doi.org/10.1038/s42003-026-09702-y.

