## [Transparent Peer Review File · Communications Biology]

Sex-dependent rescue of memory and synaptic deficits in AD model mice by increasing PSD-95 palmitoylation

Corresponding Author: Professor Kim Dore

Version 0:

Reviewer comments:

Reviewer #1

(Remarks to the Author)

In this study, Du et al. examined the sex-specific role of PSD-95 palmitoylation in synaptic function and memory by using an APP/PS1 Alzheimer's disease (AD) model. The authors demonstrated that the levels of palmitoylated PSD-95 differ between male and female mice. Furthermore, they showed that inhibiting depalmitoylation rescues cognitive and synaptic deficits in female APP/PS1 mice, but not in male mice. Given the known sex differences in AD incidence, the inclusion of sex-specific analyses is particularly interesting and valuable. However, much of the presented data does not sufficiently support or convincingly demonstrate this idea. The study requires major revisions before it can be considered for publication.

Major concerns:

1. The main concern with this study is that Palm B lacks the specificity to conclude that the memory and synaptic deficits observed in APP/PS1 mice are rescued by increased palmitoylation of PSD-95. Palm B is not specific to PSD-95 and depalmitoylates other proteins as well. To determine if the rescue effects are mediated by PSD-95 palmitoylation, this study should additionally investigate depalmitoylating enzymes specific to PSD-95, such as ABHD17. Otherwise, the focus of the study may need to shift from PSD-95 palmitoylation to general palmitoylation, stating, "Sex-dependent rescue of memory and synaptic deficits in AD model mice by increasing palmitoylation."
2. The manuscript focuses on the female AD model. However, the introduction does not sufficiently explain the rationale for comparing males and females or the reason for focusing specifically on females in this study. An additional paragraph describing the prevalence and clinical importance of AD in females, as well as previously reported sex-specific synaptic or behavioral features observed in female AD models compared to males, is recommended.
3. For the second highlight on page 2, it would be advisable to add "in female APP/PS1 mice" after "Memory impairments."
4. Figure S1 should include male mouse data.
5. Please include the ABE assay for females and males of WT and APP/PS1 mice in Figure 1. It may be worthwhile to perform an additional ABE assay as a conventional approach to validate and cross-check the findings.
6. Figures 1E and 1F show PF11 labeling of palmitoylated PSD-95 in green and PSD-95 detected by the antibody in red. However, when the two images are merged, many PF11 signals do not overlap with PSD-95 signals, raising concerns. It is possible that PF11 does not exclusively detect palmitoylated PSD-95, but could also bind to other palmitoylated proteins. Therefore, it is important to verify that PF11 specifically labels palmitoylated PSD-95.
7. In lines 106-108, it would be more appropriate to specify the age, sex, or experimental methods referenced in the earlier reported studies to make the sentence more coherent and convincing.
8. Please include the information on which types of statistics were applied to each data set in the legends.
9. In line 128, Figure S2A was not properly cited. Please include the description of the results of Figure S2A.

10. In Figures 2A-2E and supplementary Figure 4S, please include quantification data on the latency to the platform (zone) for training days 1 to 7 training and the probe test.

11. (1) From a methodological perspective, the water in the Morris Water Maze appears insufficiently opaque, which raises concerns that the hidden platform might be visible during the invisible learning phase, potentially confounding the results. (2) It is unclear why a mixed design incorporating both visible and invisible learning phases was used prior to the probe test. Clarifying the rationale behind this experimental strategy would be helpful. (3) The timeline in Figure 2A indicating the periods of visible and invisible learning is not clearly marked. This should be revised for clarity.

12. Considering that palmitic acid cannot cross the blood-brain barrier, the level of palmitic acid in the hippocampus may be more relevant than its plasma concentration. Therefore, it would be meaningful to quantify and compare the levels of palmitic acid and glyceryl palmitate within the hippocampus.

13. Please include representative images of the Palm B-treated group alongside Figures 4A and 4B for proper comparison.

14. It would be wondered if there are differences in spine density, size, and number between females and males. Please include those data in Figure 5.

15. In lines 365-366, it is difficult to describe as "it specifically targets smaller spines." It would give more information if the authors perform an analysis of the Palm B effect per spine size in addition to Figure 5J.

16. In Figure 5B, please present representative raw and scaled traces of mEPSCs recorded from all four groups (e.g., WT and APP/PS1 with or without PB treatment). Including these traces would enable a clearer, more comprehensive comparison.

17. Including a discussion of the potential mechanisms responsible for the female-specific role of palmitoylated PSD-95 in synaptic and cognitive function would improve the manuscript.

Minor concerns:

1. Regarding the keywords on page 2, it is questioned whether the words "ABHD17" and "MAGUK" are appropriate keywords for this study.

2. Please insert a space between numerical values and their units throughout the manuscript. For example, "10mg/kg" on line 73 should be revised to "10 mg/kg". Please make similar corrections throughout the manuscript.

3. Please spell out all abbreviations in full upon their first use in the manuscript. For example, IHC, ABHD17, and so on.

4. In the IHC image quantification results, does N=17 (line 132) or N=24 (line 132) refer to the number of individual mice? Please clarify what N represents in the main text (line 132) and in Figure 1G (lines 154-155) as well as in the figure legends for Figures 2D, 2E, 2L, and 2M, and for Supplementary Figures S2B, S3A, S3B, S5E, and S5F.

5. Figure 2F labels the molecular weight of actin as 35 kDa. However, it is known to be approximately 42 kDa. Please verify this label and correct it if necessary.

6. The meaning of the signal represented on the Y-axis in Figures 3B, 3C, and S6 is unclear. Please revise the Y-axis titles to be more specific and appropriate. In addition, please add a Y-axis title to Figure 3A.

7. The contrast of the GFAP IHC images in Figures 4A and 4B appears to differ between groups. Since this figure is intended for comparison, it should present images with consistent contrast settings.

8. Some reagents are listed with the manufacturer's name and detailed catalog number throughout the Materials and Methods section, while others are not. Please provide the catalog numbers for all reagents mentioned.

9. Please keep consistency in word usage throughout the manuscript, including figures and their legends: e.g., Western Blot vs Western blot.

Reviewer #2

(Remarks to the Author)

In the manuscript "Sex-dependent rescue of memory and synaptic deficits in AD model mice by increasing PSD-95 palmitoylation", Du et al. investigate the sex-dependent effects of protein thioesterase inhibitor Palmostatin B on synaptic and cognitive function in an experimental model of Alzheimer's disease. The authors used APP/PS1 mice and they evaluated the levels of PSD95 palmitoylation. They also intraperitoneally treated mice with Palmostatin B and investigated the effects on dendritic spines, synaptic function and memory. Specifically, the authors reported that: 1) PSD-95 palmitoylation is reduced in the hippocampus of APP/PS1 female mice; 2)

Palmostatin B administration counteracts memory deficits and restores PSD-95 palmitoylation in APP/PS1 female mice; 3) Palmostatin B reverts alterations of both synaptic transmission and dendritic spines in female APP/PS1 mice.

Overall, the topic dealt with in this paper is interesting, but the design of experiments and the interpretation of the results are not convincing. The main criticism is a biased analysis of APP/PS1 mice focalized only on PSD95 palmitoylation, whereas the targets of thioesterase enzymes and eventually of Palmostatin B may be very numerous.

Below are the main critical observations.

- 1) The quality of IHC images should be significantly improved. In Fukata et al. (doi: 10.1083/jcb.201302071) the hippocampus hybridization with PF11 antibody showed a staining pattern consistent with that of a conventional PSD-95 antibody. The authors should verify the protocol used for IHC and explain the differences in IHC detection signals between PF11 and total PSD95 antibodies (it seems that PF11 detects signals in cells that are not PSD95 immunoreactive).
- 2) Is there any regional difference in PF11 immunoreactivity within the hippocampus of APP/PS1 mice?
- 3) It would be also useful to provide the time spent in all quadrants during the probe test of Morris water maze to examine if all experimental groups show statistically significant differences between the time spent in the target quadrant and the time spent in all other quadrants.
- 4) How were the levels of total protein S-palmitoylation in the hippocampus of female APP/PS1 mice? In a recent paper Natale et al. (doi: 10.1073/pnas.2402604121) found elevated levels of total protein S-palmitoylation in the hippocampus of another AD experimental model (i.e., 3xTg-AD mice). The authors should at least investigate the S-palmitoylation levels of other synaptic palmitoylated proteins (e.g., GluA1, GluA2, GRIN2A, GRIN2B) to provide a more complete picture of palmitoylation levels at the post-synaptic density area. These experiments should be performed at rest and after PalmB administration.
- 4) Protein S-palmitoylation is a very dynamic post-translational modification targeting a large number of synaptic and non-synaptic proteins. In some cases (e.g., PSD95), S-palmitoylation promotes the recruitment of protein in synapse, whereas in some other (e.g., GluA1, GluA2) it inhibits the synaptic localization of targets. How do the authors explain the selective activity of PalmB on PSD95?
- 5) PalmB is an inhibitor of protein thioesterase APT1, APT2 and ABHD17, which targets many proteins in addition to PSD95. Did the authors verify the S-palmitoylation levels of other APT1 targets?
- 6) The authors reported reduced levels of palmitic acid and precursor glyceryl palmitate in the plasma of female APP/PS1 mice and suggest that "lipid differences in female APP/PS1 mice might affect palmitoylation in the brain and consequently change levels of synaptic PSD-95". However, the levels of brain palmitic acid are regulated from diet, local de novo lipogenesis (doi: 10.1111/jnc.15539) or hepatic lipogenesis (<https://doi.org/10.1038/s41467-023-44388-4>). Can the authors analyze the expression levels of main de novo lipogenesis enzymes in both hippocampus and liver of female WT and APP/PS1 mice?
How the authors explain the effects of PalmB on the levels of palmitic acid? The drug increases the levels of protein S-palmitoylation but also the levels of its substrate. How can it work?
- 7) It sounds very unusual to still study the amyloid plaques (and use the Thioflavin-S staining) in the field of AD. It is now widely accepted that amyloid plaques do not have any relationship with AD pathology and there are many more specific tools to investigate the A β production (e.g., ELISA assay, D54D2 antibody).
- 8) How the authors explain the different effect of PalmB on dendritic spine density of WT and APP/PS1 mice (Fig. 5I)?
- 9) IHC analyses must be carried out on at least 3 animals per experimental group (with more than 5 brain slices per animal). How have the statistical analyses been performed? They should be performed on animals, not on slices or ROI.

Minor:

- 1) Please, specify in the figure legends if the n of IHC refers to animals, slices or ROI
- 2) Many reports highlighted the critical role of high protein S-palmitoylation in AD-like phenotype (e.g., doi: 10.1073/pnas.1708568114 ; doi: 10.1016/j.celrep.2021.109134) but the authors did not discuss their evidence in light of these results.
- 3) The criteria for data exclusion or inclusion should be detailed in materials and methods section.

Reviewer #3

(Remarks to the Author)

I co-reviewed this manuscript with one of the reviewers who provided the listed reports. This is part of the Communications Biology initiative to facilitate training in peer review and to provide appropriate recognition for Early Career Researchers who co-review manuscripts.

Reviewer #4

(Remarks to the Author)

Review of the manuscript by Du et al.

The palmitoylation of PSD-95 has been a focal point of molecular neuroscience research, particularly in the context of synaptic function in healthy neurons. However, its contribution under pathological conditions such as Alzheimer's disease (AD) remains underexplored. In this manuscript, Du and colleagues present compelling data suggesting that modulation of PSD-95 palmitoylation could have therapeutic implications for AD, specifically in female mice.

Through a combination of the APEG assay and immunohistochemical labeling, the study reveals a selective reduction in

PSD-95 palmitoylation in the hippocampus of female APP/PS1 mice. This molecular change correlates with lower systemic palmitate concentrations, potentially restricting the pool of available substrate for protein palmitoylation. Intriguingly, systemic delivery of Palmostatin B—a small molecule inhibitor of depalmitoylating enzymes—successfully restores PSD-95 palmitoylation in this AD mouse model.

A key strength of the study lies in demonstrating that peripheral administration of Palmostatin B can influence brain palmitoylation status, supporting its relevance as a tool for modulating lipid-based posttranslational modifications in neurodegenerative diseases. This finding opens the door to targeting palmitoylation pharmacologically in AD and related disorders.

Overall, the study is well-designed, and the data are persuasive. Nonetheless, several aspects of the manuscript would benefit from additional clarification and refinement to increase its scientific rigor and accessibility.

Major Comments

1. The APEG approach offers enhanced specificity compared to ABE method. The authors' efforts to quantify mono- and di-palmitoylated PSD-95 species are commendable. However, a more granular analysis stratified by sex and genotype would strengthen the mechanistic interpretation. In particular, comparing the relative abundance of singly versus doubly palmitoylated forms between male and female mice could uncover sex-dependent regulation and potential compensatory mechanisms.

2. The dendritic spine image included in the supplementary material is of notably poor quality. It lacks the resolution typical of confocal microscopy and appears to have been captured using widefield or epifluorescence techniques. Moreover, the dendritic shafts are swollen, which raises concerns about the health of the neurons being analyzed. The authors should consider replacing this image with higher-quality data and clarifying the imaging method used.

Additionally, in Figure 5G–H, the figure legend omits a scale bar for the spine images, which must be included for accurate spatial interpretation.

3. The manuscript would benefit from a more thorough introduction to the topic of protein palmitoylation. A clearer explanation of S-palmitoylation, where palmitic acid is covalently attached to cysteine residues via thioester linkage, would be particularly useful for readers unfamiliar with lipid modifications. It is also important to distinguish S-palmitoylation from less common lipid modifications such as O-palmitoylation (to serine or threonine residues) and N-palmitoylation. Additionally, the manuscript should acknowledge the current shift in terminology toward the broader term S-acylation, which refers to the reversible attachment of fatty acids to cysteine residues via thioester bonds. While S-palmitoylation (attachment of palmitic acid) is the most prevalent and well-characterized form, commonly used detection methods do not differentiate between different fatty acids. Thus, modifications such as S-stearoylation, S-oleoylation, or S-myristoleoylation may also be present but indistinguishable using standard techniques. Including this clarification in the Introduction would improve scientific precision and reflect current understanding in the field.

Certain claims in the Introduction should be revised for accuracy. For example, while S-palmitoylation is considered a dynamic and reversible lipid modification, it is not the only one that is reversible, nor can it be definitively labeled as the most prevalent across all systems. Similarly, the statement that "40–50% of synaptic proteins are palmitoylated" is speculative. Proteomic studies using synaptoneurosomes and mass spectrometry suggest a lower proportion (more likely in the range of 10–15% given current detection limits).

The text on page 3 introduces a study with "In a recent study..." but authors do not provide a corresponding reference. This should be corrected.

4. While Palmostatin B rescues PSD-95 palmitoylation in female APP/PS1 mice, its broader biological effects remain insufficiently characterized. Notably, wild-type and male APP/PS1 mice do not show similar changes upon treatment. This raises the question of whether the observed effects are specific to enzyme inhibition or secondary to altered systemic palmitate levels. The authors should discuss or test whether the observed rescue of PSD-95 palmitoylation is mechanistically linked to PPT1 inhibition, or whether other pathways might contribute.

Additionally, it is plausible that other palmitoylated proteins are also affected by Palmostatin B. This is particularly important since widespread increases in protein palmitoylation have been implicated in neurotoxicity, as seen in PPT1 deficiency disorders. Moreover, elevated palmitate levels (e.g., from high-fat diets) are known to impair cognitive function. These caveats should be discussed, especially in light of the fact that proteins such as GluA1 (AMPA subunit), which require dynamic palmitoylation for synaptic plasticity, were not evaluated in this study.

Minor Comments

In the dendritic spine analysis, while the number of analyzed dendritic segments is specified, the number of biological replicates (animals) is not. This information is essential for assessing statistical power. For spine data where multiple segments come from the same animal, nested ANOVA is the appropriate statistical method and should be applied or justified otherwise.

Version 1:

Reviewer comments:

Reviewer #1

(Remarks to the Author)

The authors addressed most of my concerns, and the manuscript has been improved. However, I still have a few major concerns.

Major concerns:

1. Despite the authors' response to my previous concern, there is still concern about the specificity of the PF11 antibody used to label palmitoylated PSD-95. The original paper using the PF11 antibody (Fukata et al., 2013, JCB) shows convincing overlap between PF11 and total PSD-95. Please refer to Figure 1E of the Fukata et al., 2013, JCB paper for images. It appears that the PF11 antibody used in this study (#AG-27B-0021-C100, AdipoGen) is ineffective. Are there any publications using this PF11 antibody from AdipoGen? The authors should replace Figures 1I-1K and 2H-2K with data obtained using a functional PF11 antibody. The current PF11 images are not suitable for publication.

2. The experimental sequence of the Morris water maze in Figure 2A is somewhat unusual. Visible learning, a stronger stimulus for investigating egocentric learning and memory, was performed first (days 1-3). Then, the experiment progressed by gradually removing the flag (days 4-5) or making the platform invisible (days 6-7), which change the intensity and type of training. This experimental protocol differs from those described in the two referenced papers (refs #8 and #87) in the Methods section.

The rebuttal to 11-(2) is somewhat understandable. However, performing invisible learning after visible learning to confirm spatial memory complicates the interpretation, because it may reflect memory retention for egocentric learning and memory. The experimental protocol, combining invisible learning after visible learning, seems to be a more specialized method for examining egocentric memory retention or memory recall. Have there been any reports of a significant decline in memory retention or recall in female APP/PS1 mice or female AD patients? If so, this could be interpreted more convincingly. Please add this information.

From another perspective, these invisible and visible learning experiments are typically conducted separately in spatial learning research because they have different evaluation purpose. Please consider this and verify the experimental sequence and its interpretation. In addition, please add the reference from Zhang et al. (2012) to the reference section.

3. The authors provided new data showing the palmitoylation of total proteins in Figures 1A and 1B. Please expand the upper immunoblot image in Figure 1B to include the proteins around 245 kDa, as shown in Figure 1A.

Minor concerns:

1. The (V) and (PB) labels in Figures 2H and 2I are incorrect. The group labels should be on the left side of the images. Please correct them.

2. In the Y-axis of Figure 2J, (N to WT) may need to be corrected to (N to V).

Reviewer #2

(Remarks to the Author)

The authors addressed all the Reviewer's concerns.

Reviewer #3

(Remarks to the Author)

I co-reviewed this manuscript with one of the reviewers who provided the listed reports. This is part of the Communications Biology initiative to facilitate training in peer review and to provide appropriate recognition for Early Career Researchers who co-review manuscripts.

Reviewer #4

(Remarks to the Author)

I have reviewed the revised manuscript, and I am pleased with the authors' thorough and thoughtful responses to my previous comments. All concerns have been fully and satisfactorily addressed, and the revisions have significantly improved the clarity and overall quality of the work.

I have no further suggestions for the authors at this stage. The manuscript is well written, scientifically sound, and suitable for publication in its current form.

I therefore recommend acceptance of the manuscript.

Version 2:

Reviewer comments:

Reviewer #1

(Remarks to the Author)

The authors responded to the points raised in the second revision, but some issues remain unresolved.

1. Regarding the concerns about the specificity of the PF11 antibody, the ICC images provided by the authors in their response 1-E further confirm that the PF11 antibody used in this study is not specific to palmitoylated PSD-95.

In response 1-E, the images show that the antibody used in Chowdury and Hell (2019) worked well in ICC, but the antibody used in this study did not work, even in ICC. Since the Chowdury and Hell (2019) study used the PF11 antibody obtained from the Fukata lab, I recommend that the authors obtain the PF11 antibody from the Fukata lab and replace the Figures 1I-1K and 2H-2K. In my opinion, AdipoGen's PF11 antibody is not suitable for IHC or ICC, whereas the PF11 antibody from the Fukata lab works for both.

I agree with the author's main point that palmitoylated PSD-95 levels are lower in the hippocampus of female APP/PS1 mice. However, the data presented in Figures 1I-1K and 2H-2K, which were labeled by AdipoGen's PF11 antibody, are unreliable.

2. Regarding the Morris water maze experiments, the replacement of the references has been appropriately addressed. However, the previous comment was not only about the references themselves, but also about the need for an appropriate interpretation of the unusual experimental protocol used in the Morris water maze test in this study. Since this protocol differs from conventional designs, a tailored interpretation is necessary. Unfortunately, this aspect has not been addressed in the revised version.

I encourage the authors to provide a clear discussion that interprets the behavioral results in the context of this modified experimental protocol. In addition, have there been previous reports indicating impaired memory recall or retention specifically in females? Including a discussion of such prior findings would provide a novel perspective and enrich the manuscript further.

Version 4:

Reviewer comments:

Reviewer #1

(Remarks to the Author)

I would like to thank the editorial team for their great compromise.

Regarding line 702 (or 761 in the marked-up version), where the authors refer to the Fukata paper "as in the original publication" in relation to their IHC optimization, I would suggest removing "as in the original publication," since Fukata's and AdipoGen's antibodies, despite having the same name, are different.

The following should be sufficient:

"we performed an experiment using 8 μ m thin sections fixed in acetone⁸⁴"

We thank the reviewers for their comprehensive assessment of our manuscript and their numerous comments. To address the concerns raised by the reviewers, we substantially revised our manuscript and included several new experiments, analyses and discussions. Please see our point-by-point answers below. For clarity, all reviewers' comments are *italicized* and our responses are not.

Reviewer #1

In this study, Du et al. examined the sex-specific role of PSD-95 palmitoylation in synaptic function and memory by using an APP/PS1 Alzheimer's disease (AD) model. The authors demonstrated that the levels of palmitoylated PSD-95 differ between male and female mice. Furthermore, they showed that inhibiting depalmitoylation rescues cognitive and synaptic deficits in female APP/PS1 mice, but not in male mice. Given the known sex differences in AD incidence, the inclusion of sex-specific analyses is particularly interesting and valuable. However, much of the presented data does not sufficiently support or convincingly demonstrate this idea. The study requires major revisions before it can be considered for publication.

Major concerns:

1. The main concern with this study is that Palm B lacks the specificity to conclude that the memory and synaptic deficits observed in APP/PS1 mice are rescued by increased palmitoylation of PSD-95. Palm B is not specific to PSD-95 and depalmitoylates other proteins as well. To determine if the rescue effects are mediated by PSD-95 palmitoylation, this study should additionally investigate depalmitoylating enzymes specific to PSD-95, such as ABHD17. Otherwise, the focus of the study may need to shift from PSD-95 palmitoylation to general palmitoylation, stating, "Sex-dependent rescue of memory and synaptic deficits in AD model mice by increasing palmitoylation."

We agree with the reviewer that a significant limitation of our study is the fact that Palm B is not specific to PSD-95. To acknowledge this from the start of the manuscript, we changed the wording of the sentence introducing Palm B, see line 22. The reviewer's suggestion of manipulating the expression of ABHD17 could be an interesting follow-up but would require extensive new experiments that are outside the scope of this study. We attempted to measure ABHD17a in our animals for several weeks to assess if its expression changed in APP/PS1 mice but could not get good signal with the antibody we tried (15854-1-AP from Proteintech, cited in Yokoi et al 2016). To provide more information about the specificity of Palm B, we performed an additional experiment and measured if palmitoylation of p62, a major target of APT1, another enzyme inhibited by Palm B. We found no differences in p62 palmitoylation in Palm B injected animals, indicating that Palm B does not affect its palmitoylation, these results are now shown in Figure S8.

2. The manuscript focuses on the female AD model. However, the introduction does not sufficiently explain the rationale for comparing males and females or the reason for focusing specifically on females in this study. An additional paragraph describing the prevalence and clinical importance of AD in females, as well as previously reported sex-specific synaptic or behavioral features observed in female AD models compared to males, is recommended.

We thank the reviewer for this suggestion. See lines 44-55 in the revised manuscript.

3. For the second highlight on page 2, it would be advisable to add "in female APP/PS1 mice" after "Memory impairments."

We agree with the reviewer that it would be better to specify that the memory impairments were rescued specifically in female APP/PS1 mice. However, there is a character limit for highlights, which is why it was written as is.

4. Figure S1 should include male mouse data.

The purpose of the data shown in the previous Figure S1 (now Figure S2) was to demonstrate that the reduction in PSD-95 palmitoylation observed in female APP/PS1 mice is specific to the hippocampus. We found that in the frontal cortex, PSD-95 palmitoylation was similar in WT and APP/PS1 female mice. Since we did not observe any reductions of PSD-95 palmitoylation in the hippocampus of male APP/PS1 mice, it is very unlikely that any changes in PSD-95 palmitoylation would be observed in the frontal cortex of male mice. We also now provide new data showing that global palmitoylation, detected with an antibody binding to all palmitoylated cysteines, is also reduced in the hippocampus of female APP/PS1 mice, with no effects in male mice (Figure 1A-D). Therefore, we believe that measuring PSD-95 palmitoylation in the frontal cortex of male mice would not bring anything interesting to this study and decided not to perform these additional experiments.

5. Please include the ABE assay for females and males of WT and APP/PS1 mice in Figure 1. It may be worthwhile to perform an additional ABE assay as a conventional approach to validate and cross-check the findings.

Both the ABE and the APEGS assay have been used by several different groups to measure protein palmitoylation. For example, Dell'Acqua lab used the APEGS assay to measure AKAP150 palmitoylation (doi: 10.1016/j.celrep.2018.09.085), as well as Xu lab investigated IFITM proteins palmitoylation by ABHD16A (<https://doi.org/10.1128/mbio.02289-22>). Therefore, we don't see why a validation of our results is needed by using a very similar biochemical method. We do confirm and validate our results using a different method: IHC using the PF11 antibody. Moreover, as noted above, we now provide additional data showing a reduction of total protein palmitoylation in female APP/PS1 mice (Figure 1A-D).

6. Figures 1E and 1F show PF11 labeling of palmitoylated PSD-95 in green and PSD-95 detected by the antibody in red. However, when the two images are merged, many PF11 signals do not overlap with PSD-95 signals, raising concerns. It is possible that PF11 does not exclusively detect palmitoylated PSD-95, but could also bind to other palmitoylated proteins. Therefore, it is important to verify that PF11 specifically labels palmitoylated PSD-95.

We agree with the reviewer that the overlap is not perfect. To confirm that the PF11 antibody is specific for palmitoylated PSD-95, we performed experiments in PSD-95 KO mice and saw no signal with both the PSD-95 and PF11 antibodies (Figure S3, formerly Figure S2). Because there was no signal with the PF11 antibody in slices from PSD-95 KO mice, it is very unlikely that this antibody binds to other palmitoylated proteins.

7. In lines 106-108, it would be more appropriate to specify the age, sex, or experimental methods referenced in the earlier reported studies to make the sentence more coherent and convincing.

We thank the reviewer for this comment and changed the sentence to include more details. See lines 136-138 for the improved sentence.

8. Please include the information on which types of statistics were applied to each data set in the legends.

We now added a description of all statistical tests used in all figure legends.

9. In line 128, Figure S2A was not improperly cited. Please include the description of the results of Figure S2A.

We are sorry for this lack of details. We now describe the results shown in Figure S3A (formerly Fig. S2A) in lines 163-165.

10. In Figures 2A-2E and supplementary Figure 4S, please include quantification data on the latency to the platform (zone) for training days 1 to 7 training and the probe test.

We included latency to platform data in Figure S6 for training days 1-7. Since the platform zone is small, several animals don't enter this zone during the probe test; therefore, the data looking into latency to the platform is not really informative for the probe test. See below the latency to first entry in Zone 1 (quadrant where the platform was located) during the probe test. In our opinion, it does not bring more information than the % time in Zone 1 shown in Figure 2, therefore we decided to not include it in Figure S6. Please note that more supplementary data for the Morris Water Maze test were requested by other reviewers, these can be found in Figure S6 as well.

11. (1) From a methodological perspective, the water in the Morris Water Maze appears insufficiently opaque, which raises concerns that the hidden platform might be visible during the invisible learning phase, potentially confounding the results.

We were trained on the Morris Water Maze test by the Rissman lab who are experts in this behavioral test (see reference Zhang et al, 2012). The picture was taken during the visible-platform phase of testing. We confirm that during the hidden-platform phase, the platform was obscured, as the white tempera reliably made it invisible to the mice.

(2) It is unclear why a mixed design incorporating both visible and invisible learning phases was used prior to the probe test. Clarifying the rationale behind this experimental strategy would be helpful.

Most publications reporting Morris Water Maze data include both phases of the task; for example, see Zhang et al, 2012 and <https://doi.org/10.1038/s41598-022-09270-1>. The visible-platform phase provides an easier task for the mice and serves to rule out potential motor or vision deficits, as mice that fail to learn the location of the visible platform might have such deficits. The invisible-platform phase directly assesses spatial memory.

(3) The timeline in Figure 2A indicating the periods of visible and invisible learning is not clearly marked. This should be revised for clarity.

We thank the reviewer for pointing that out. See the new clarified timeline in Figure 2A.

12. Considering that palmitic acid cannot cross the blood-brain barrier, the level of palmitic acid in the hippocampus may be more relevant than its plasma concentration. Therefore, it would be meaningful to quantify and compare the levels of palmitic acid and glyceryl palmitate within the hippocampus.

Palmitic acid from diet does cross the blood-brain barrier, a convincing demonstration can be found in this paper: <https://doi.org/10.3389/fnins.2018.00552> which found that palmitic acid supplementation increased brain palmitic acid and increased palmitoylation of two proteins in the brain. This other paper: doi: 10.1111/jnc.15539 also confirm that palmitic acid supplementation from diet increases brain levels. Nevertheless, we did perform additional experiments and measured palmitic acid and other free fatty acids in hippocampal tissue to address another reviewer's comments, these results are now included in Figure 3 (see also lines 298-313 for their description).

13. Please include representative images of the Palm B-treated group alongside Figures 4A and 4B for proper comparison.

We thank the reviewer for this suggestion, we added these images and because we needed to reimage the Palm B treated slices to acquire high magnification pictures (insets), we reimaged a representative example for all the conditions so that all images were acquired using the same settings.

14. It would be wondered if there are differences in spine density, size, and number between females and males. Please include those data in Figure 5.

We agree with the reviewer that this is an interesting question. To answer it, we performed an additional experiment in which we stained acute brain slices from WT and APP/PS1 male mice with Dil, using the same protocol applied to female mice. After imaging and analysis, we compared spine size and density between male and female mice, and these results are shown in Figure S12. Consistent with our findings in female mice, we found that spine size also increased in male APP/PS1 mice compared to WT (Fig. S12C). Interestingly, dendritic spines of both WT and APP/PS1 male mice were significantly smaller than those in female mice, consistent with previous literature (see <https://www.nature.com/articles/s41598-020-68371-x> and <https://www.nature.com/articles/s41598-024-62951-x>). Moreover, spine density was similarly reduced in male and female APP/PS1 mice (Fig. S12D). To determine whether the reduction in spine density was driven by changes in small or large spines, we calculated the median spine size for male and female WT mice, classified different size of spines accordingly, and evaluated the density of small and large dendritic spines respectively. We found similar trends in both sexes, but effect size and significance of results were larger in female mice (Fig. S12E,F).

15. In lines 365-366, it is difficult to describe as “it specifically targets smaller spines.” It would give more information if the authors perform an analysis of the Palm B effect per spine size in addition to Figure 5J.

To support our claim that Palm B specifically targets small spines, we performed an additional analysis of our Dil staining images as described above at point 14. We thank the reviewer for this suggestion, as it reinforces our conclusion that small dendritic spines are preferentially affected in APP/PS1 mice and that Palm B treatment restores those vulnerable dendritic spines.

16. In Figure 5B, please present representative raw and scaled traces of mEPSCs recorded from all four groups (e.g., WT and APP/PS1 with or without PB treatment). Including these traces would enable a clearer, more comprehensive comparison.

Representative traces from all groups are now included in Figure 5B.

17. Including a discussion of the potential mechanisms responsible for the female-specific role of palmitoylated PSD-95 in synaptic and cognitive function would improve the manuscript.

We added some text in the discussion about this at lines 511-518 about possible mechanisms for this sex difference. We also think that differences in NMDAR signaling are also involved (see lines 520-540).

Minor concerns:

1. Regarding the keywords on page 2, it is questioned whether the words “ABHD17” and “MAGUK” are appropriate keywords for this study.

We respectfully disagree, in this manuscript we present substantial data showing that PSD-95 (a MAGUK) palmitoylation is reduced in the hippocampus of female APP/PS1 mice and that treatment with Palm B, which is a potent inhibitor of ABHD17 enzymes (as well as other depalmitoylating enzymes) rescues PSD-95 palmitoylation.

2. Please insert a space between numerical values and their units throughout the manuscript. For example, “10mg/kg” on line 73 should be revised to “10 mg/kg”. Please make similar corrections throughout the manuscript.

We thank the reviewer for pointing that out. We have now inserted spaces between all numerical values and their units throughout the manuscript.

3. Please spell out all abbreviations in full upon their first use in the manuscript. For example, IHC, ABHD17, and so on.

This has been corrected, see lines 87 and 159.

4. In the IHC image quantification results, does N=17 (line 132) or N=24 (line 132) refer to the number of individual mice? Please clarify what N represents in the main text (line 132) and in Figure 1G (lines 154-

155) as well as in the figure legends for Figures 2D, 2E, 2L, and 2M, and for Supplementary Figures S2B, S3A, S3B, S5E, and S5F.

Yes, all the 'N' for IHC and most of the data shown in this paper are indicating individual mice. We now specify this at line 174 in the main text and in Fig. 1 legend. We specified that 'N' denotes the number of mice in Figure 2 and all supplementary figures with IHC data.

5. Figure 2F labels the molecular weight of actin as 35 kDa. However, it is known to be approximately 42 kDa. Please verify this label and correct it if necessary.

We thank the reviewer for pointing that out, we revised the label accordingly and verified all other Western Blots molecular weight markers.

6. The meaning of the signal represented on the Y-axis in Figures 3B, 3C, and S6 is unclear. Please revise the Y-axis titles to be more specific and appropriate. In addition, please add a Y-axis title to Figure 3A.

The signal in these data is obtained from the chromatograms as shown in Figure 3A, we now indicate that the units are in 'Counts' as detected with the GC/MS instrument.

7. The contrast of the GFAP IHC images in Figures 4A and 4B appears to differ between groups. Since this figure is intended for comparison, it should present images with consistent contrast settings.

We take image presentation very seriously. For all images presented in this manuscript, we ensured that the brightness and contrast settings were identical for groups within the same figure.

8. Some reagents are listed with the manufacturer's name and detailed catalog number throughout the Materials and Methods section, while others are not. Please provide the catalog numbers for all reagents mentioned.

Thank you for pointing this out. We revised the methods section accordingly.

9. Please keep consistency in word usage throughout the manuscript, including figures and their legends: e.g., Western Blot vs Western blot.

We thank the reviewer for pointing that out. We have now ensured consistent terminology throughout the manuscript and figures.

Reviewer #2

In the manuscript "Sex-dependent rescue of memory and synaptic deficits in AD model mice by increasing PSD-95 palmitoylation", Du et al. investigate the sex-dependent effects of protein thioesterase inhibitor Palmostatin B on synaptic and cognitive function in an experimental model of Alzheimer's disease. The authors used APP/PS1 mice and they evaluated the levels of PSD95 palmitoylation. They also intraperitoneally treated mice with Palmostatin B and investigated the effects on dendritic spines,

synaptic function and memory. Specifically, the authors reported that: 1) PSD-95 palmitoylation is reduced in the hippocampus of APP/PS1 female mice; 2) Palmostatin B administration counteracts memory deficits and restores PSD-95 palmitoylation in APP/PS1 female mice; 3) Palmostatin B reverts alterations of both synaptic transmission and dendritic spines in female APP/PS1 mice. Overall, the topic dealt with in this paper is interesting, but the design of experiments and the interpretation of the results are not convincing. The main criticisms is a biased analysis of APP/PS1 mice focalized only on PSD95 palmitoylation, whereas the targets of thioesterase enzymes and eventually of Palmostatin B may be very numerous.

Below are the main critical observations.

1) The quality of IHC images should be significantly improved. In Fukata et al. (doi: 10.1083/jcb.201302071) the hippocampus hybridization with PF11 antibody showed a staining pattern consistent with that of a conventional PSD-95 antibody. The authors should verify the protocol used for IHC and explain the differences in IHC detection signals between PF11 and total PSD95 antibodies (it seems that PF11 detects signals in cells that are not PSD95 immunoreactive).

We agree with the reviewer that our images differ from the ones shown in the 2013 paper by the Fukata group. We note that most of the images shown in this 2013 publication were obtained from primary hippocampal neurons and not in tissue. Nevertheless, we decided to do an additional experiment using the exact same protocol as the Fukata group. We obtained the protocol from Dr. Masaki Fukata, using thin 8 µm slices cut with a cryostat and fixed with acetone. Using this protocol, we were able to observe only punctate signal with the PF11 antibody and no signal in the cell bodies as shown in new Figure S3C. In these samples, we also saw a significant reduction in PF11 signal in female APP/PS1 mice, confirming our results obtained with PFA fixed, 50 µm thick slices. Results obtained in PSD-95 KO mice (Fig. S3A) also demonstrate the specificity of the PF11 antibody. Together, these data confirm that the PF11 antibody can measure PSD-95 palmitoylation in more conventional samples. See lines 173-176 for a description of these results in the main text.

2) Is there any regional difference in PF11 immunoreactivity within the hippocampus of APP/PS1 mice?

We thank the reviewer for this suggestion. We imaged PF11 and PSD-95 immunoreactivity (as well as MAP2) in the dentate gyrus and the subiculum sub-regions of the hippocampus. Interestingly, we found that both PF11 and PSD-95 was significantly reduced in the subiculum of female APP/PS1 mice. This was not observed in male mice. Moreover, no significant differences were found in the dentate gyrus for both female and male mice. These results are now included in Figure S4 and discussed in lines 177-180 of the manuscript.

3) It would be also useful to provide the time spent in all quadrants during the probe test of Morris water maze to examine if all experimental groups show statistically significant differences between the time spent in the target quadrant and the time spent in all other quadrants.

We thank the reviewer for this suggestion. These data are now included in Figure S5. We do see a significant difference between the time spent in the target quadrant and the time spent in all other quadrants for all experimental groups, except for APP/PS1 female mice treated with vehicle.

4) How were the levels of total protein S-palmitoylation in the hippocampus of female APP/PS1 mice? In a recent paper Natale et al. (doi: 10.1073/pnas.2402604121) found elevated levels of total protein S-

palmitoylation in the hippocampus of another AD experimental model (i.e., 3xTg-AD mice). The authors should at least investigate the S-palmitoylation levels of other synaptic palmitoylated proteins (e.g., GluA1, GluA2, GRIN2A, GRIN2B) to provide a more complete picture of palmitoylation levels at the post-synaptic density area. These experiments should be performed at rest and after PalmB administration.

In the mentioned recent paper, a different mouse model and a different assay to quantify protein palmitoylation were used, which may contribute to differences in results. Moreover, their protein palmitoylation analysis combined the data from both male and female mice, thus potential sex differences could be masked. To better understand the levels of total protein palmitoylation, we performed additional experiments using an antibody detecting all palmitoylated cysteines and quantified total protein palmitoylation. We found that total protein palmitoylation was specifically reduced in the hippocampus of female APP/PS1 mice without any effects in male mice (see Figure 1A-D). We also quantified the palmitoylation of p62, a major APT1 target (see Fig. S8 and response to point 5 below). We also added text in the discussion (see lines 535-545) to acknowledge the Natale et al. paper and the implications of their findings.

4) Protein S-palmitoylation is a very dynamic post-translational modification targeting a large number of synaptic and non-synaptic proteins. In some cases (e.g., PSD95), S-palmitoylation promotes the recruitment of protein in synapse, whereas in some other (e.g., GluA1, GluA2) it inhibits the synaptic localization of targets. How do the authors explain the selective activity of PalmB on PSD95?

PSD-95 is the second most abundant protein in dendritic spines and PSD-95 palmitoylation is more dynamically regulated than other proteins (Yokoi et al 2016). We hypothesize that these are the main factors explaining the selective activity of Palm B on PSD-95. We also toned down the text all through the manuscript to address the non-specificity of Palm B.

5) PalmB is an inhibitor of protein thioesterase APT1, APT2 and ABHD17, which targets many proteins in addition to PSD95. Did the authors verify the S-palmitoylation levels of other APT1 targets?

To address this issue, we quantified the palmitoylation of the p62 protein, a major APT1 target. We found that Palm B did not affect p62 palmitoylation in female APP/PS1 mice. These results are now shown in Figure S8 and discussed at lines 249-252 in the manuscript.

6) The authors reported reduced levels of palmitic acid and precursor glyceryl palmitate in the plasma of female APP/PS1 mice and suggest that “lipid differences in female APP/PS1 mice might affect palmitoylation in the brain and consequently change levels of synaptic PSD-95”. However, the levels of brain palmitic acid are regulated from diet, local de novo lipogenesis (doi: 10.1111/jnc.15539) or hepatic lipogenesis (<https://doi.org/10.1038/s41467-023-44388-4>). Can the authors analyze the expression levels of main de novo lipogenesis enzymes in both hippocampus and liver of female WT and APP/PS1 mice?

How the authors explain the effects of PalmB on the levels of palmitic acid? The drug increases the levels of protein S-palmitoylation but also the levels of its substrate. How can it works?

To better understand how palmitic acid levels in plasma correlate with its levels in the brain, we performed additional experiments and quantified a panel of free fatty acids in hippocampal tissue. These results are now incorporated in Figure 3. We agree with the reviewer that it's confusing that Palm B increases the levels of palmitoylation substrates in mouse plasma. However, we found that in the brain, Palm B significantly reduces the proportion of palmitic and stearic acid, the main fatty acids involved in

palmitoylation, only in female APP/PS1 mice (see lines 298-313 for a description of the new lipidomics results). This supports our results indicating that Palm B only rescues PSD-95 palmitoylation and memory deficits in female APP/PS1 mice. We thank the reviewer for suggesting to analyze de novo lipogenesis enzymes, however, we believe that such an analysis is outside the scope of this study.

7) It sounds very unusual to still study the amyloid plaques (and use the Thioflavin-S staining) in the field of AD. It is now widely accepted that amyloid plaques do not have any relationship with AD pathology and there are many more specific tools to investigate the A β production (e.g., ELISA assay, D54D2 antibody).

We respectfully disagree with the reviewer. Amyloid plaques are an important feature of AD pathology. In Figure 4, we measured amyloid plaques and GFAP signal to assess if Palm B had a direct effect on these features of AD pathology. We observed no effects of Palm B on plaques or GFAP signal. Therefore, we don't see how using other tools to measure A β production would benefit our study.

8) How the authors explain the different effect of PalmB on dendritic spine density of WT and APP/PS1 mice (Fig. 5I)?

According to our new analysis of spine density separating small and large dendritic spines, the effect of Palm B in WT mice is seen only in large dendritic spines. We hypothesize that it might be due to the unspecific effects of Palm B on spines where PSD-95 palmitoylation is already maxed out, see line 448-452 and Figure 5KL.

9) IHC analyses must be carried out on at least 3 animals per experimental group (with more than 5 brain slices per animal). How have the statistical analyses been performed? They should be performed on animals, not on slices or ROI.

The N in all our IHC experiments denotes the number of animals used and is superior or equal to 7. All details about image analysis are provided in the Methods.

Minor:

1) Please, specify in the figure legends if the n of IHC refers to animals, slices or ROI.

The N in all our IHC experiments denotes the number of animals used it is now specified at line 170 and in all figure legends.

2) Many reports highlighted the critical role of high protein S-palmitoylation in AD-like phenotype (e.g., doi: 10.1073/pnas.1708568114 ; doi: 10.1016/j.celrep.2021.109134) but the authors did not discuss their evidence in light of these results.

We thank the reviewer for mentioning these studies, we now include a discussion about the possible implications of APP and BACE palmitoylation in the discussion at lines 536-545.

3) The criteria for data exclusion or inclusion should be detailed in materials and methods section.

Please see lines 714-715, 736-738, and 816-817 in the methods section for data exclusion criteria for the IHC, MWM, and mEPSCs data respectively. For all other experiments, no data points were excluded.

Reviewer #3

I co-reviewed this manuscript with one of the reviewers who provided the listed reports. This is part of the Communications Biology initiative to facilitate training in peer review and to provide appropriate recognition for Early Career Researchers who co-review manuscripts.

Reviewer #4

The palmitoylation of PSD-95 has been a focal point of molecular neuroscience research, particularly in the context of synaptic function in healthy neurons. However, its contribution under pathological conditions such as Alzheimer's disease (AD) remains underexplored. In this manuscript, Du and colleagues present compelling data suggesting that modulation of PSD-95 palmitoylation could have therapeutic implications for AD, specifically in female mice. Through a combination of the APEG assay and immunohistochemical labeling, the study reveals a selective reduction in PSD-95 palmitoylation in the hippocampus of female APP/PS1 mice. This molecular change correlates with lower systemic palmitate concentrations, potentially restricting the pool of available substrate for protein palmitoylation. Intriguingly, systemic delivery of Palmostatin B—a small molecule inhibitor of depalmitoylating enzymes—successfully restores PSD-95 palmitoylation in this AD mouse model. A key strength of the study lies in demonstrating that peripheral administration of Palmostatin B can influence brain palmitoylation status, supporting its relevance as a tool for modulating lipid-based posttranslational modifications in neurodegenerative diseases. This finding opens the door to targeting palmitoylation pharmacologically in AD and related disorders. Overall, the study is well-designed, and the data are persuasive. Nonetheless, several aspects of the manuscript would benefit from additional clarification and refinement to increase its scientific rigor and accessibility.

Major Comments

1. The APEG approach offers enhanced specificity compared to ABE method. The authors' efforts to quantify mono- and di-palmitoylated PSD-95 species are commendable. However, a more granular analysis stratified by sex and genotype would strengthen the mechanistic interpretation. In particular, comparing the relative abundance of singly versus doubly palmitoylated forms between male and female mice could uncover sex-dependent regulation and potential compensatory mechanisms.

We thank the reviewer for this suggestion, we quantified the relative abundance of non-palmitoylated, singly and doubly palmitoylated PSD-95 in all our APEGs assay results. We found that the reduction in the Palm ratio in female APP/PS1 mice is mediated by a significant increase in non-palmitoylated PSD-95 and a decrease in PSD-95 palmitoylated at 2 sites (Figure S1E). We also found that the amount of singly palmitoylated PSD-95 did not differ between WT and APP/PS1 female mice, suggesting that this PSD-95 population is stable and do not contribute to the observed changes. No such changes were observed in the frontal cortex (Figure S1B) or in male mice (Figure S1H). Interestingly, when we compared the abundance of the different palmitoylation status in APP/PS1 females treated with vehicle or Palm B (Figure 2H), we observed a reversal of the effects described above comparing WT and APP/PS1 female mice; the measured increase in Palm ratio (Figure 2I), is mediated by a decrease in non-palmitoylated PSD-95 and an increase in PSD-95 palmitoylated at 2 sites (Figure S5C). We now describe these results in lines 146-153 and 241-242.

2. *The dendritic spine image included in the supplementary material is of notably poor quality. It lacks the resolution typical of confocal microscopy and appears to have been captured using widefield or epifluorescence techniques. Moreover, the dendritic shafts are swollen, which raises concerns about the health of the neurons being analyzed. The authors should consider replacing this image with higher-quality data and clarifying the imaging method used.*

The images shown in Figure S12 (formerly S8), were obtained using a LEICA Sp8 confocal microscope, as specified in the methods (line 717). For these experiments, we used 400 μm acute brain slices, which were then fixed with PFA and stained with Dil. The relative high thickness of the slices and the fact that we wanted to show **representative unaltered** images can explain why the image quality might seem inferior. Another important point is that Dil staining in acute slices has been shown to reveal dendritic swelling (see Trivino-Paredes et al 2019; doi:10.1152/jn.00332.2019). Importantly, sections with significant swelling and/or beading were excluded from the analysis as recommended in this publication (Trivino-Paredes et al 2019; doi:10.1152/jn.00332.2019). Moreover, most of the slices used for Dil staining were obtained from the same mice used for electrophysiology experiments which strongly suggests that the slices were healthy. Similarly, in Trivino-Paredes et al 2019, acute slices from the same animals used for Dil staining were used to successfully induce synaptic plasticity.

Additionally, in Figure 5G–H, the figure legend omits a scale bar for the spine images, which must be included for accurate spatial interpretation.

The scale bar is located in the bottom right corner of the first image, all examples were imaged and are shown with the same magnification.

3. *The manuscript would benefit from a more thorough introduction to the topic of protein palmitoylation. A clearer explanation of S-palmitoylation, where palmitic acid is covalently attached to cysteine residues via thioester linkage, would be particularly useful for readers unfamiliar with lipid modifications. It is also important to distinguish S-palmitoylation from less common lipid modifications such as O-palmitoylation (to serine or threonine residues) and N-palmitoylation. Additionally, the manuscript should acknowledge the current shift in terminology toward the broader term S-acylation, which refers to the reversible attachment of fatty acids to cysteine residues via thioester bonds. While S-palmitoylation (attachment of palmitic acid) is the most prevalent and well-characterized form, commonly used detection methods do not differentiate between different fatty acids. Thus, modifications such as S-stearoylation, S-oleoylation, or S-myristoleoylation may also be present but indistinguishable using standard techniques. Including this clarification in the Introduction would improve scientific precision and reflect current understanding in the field.*

We thank the reviewer for this suggestion; this is a very important point. We have now added a section in the introduction discussing the different forms of palmitoylation, the use of the S-acylation broader term (also added as a keyword) and the fact that the detection methods used cannot distinguish between S-acylation with palmitic acid versus similar fatty acids (see lines 65-73).

Certain claims in the Introduction should be revised for accuracy. For example, while S-palmitoylation is considered a dynamic and reversible lipid modification, it is not the only one that is reversible, nor can it be definitively labeled as the most prevalent across all systems. Similarly, the statement that "40–50% of synaptic proteins are palmitoylated" is speculative. Proteomic studies using synaptoneurosomal preparations and mass spectrometry suggest a lower proportion (more likely in the range of 10–15% given current detection limits).

We modified the introduction accordingly (see lines 69-84). However, we respectfully disagree with the reviewer about the statement indicating that 40-50% of synaptic proteins are palmitoylated; the cited publications report that this estimate is accurate and was obtained specifically from proteomic and mass spectrometry studies (Sanders et al 2015; Petropavlovskiy et al, 2021). Sanders et al 2015 also report that 'data from 15 palmitoylation proteomics studies into one compendium containing 1,838 genes encoding palmitoylated proteins; representing approximately 10% of the genome', which might be where the reviewer found this lower %. We did not find publications reporting lower amounts of palmitoylated **synaptic** proteins.

The text on page 3 introduces a study with "In a recent study..." but authors do not provide a corresponding reference. This should be corrected.

We thank the reviewer for pointing that out. It is now corrected. See line 80.

4. While Palmostatin B rescues PSD-95 palmitoylation in female APP/PS1 mice, its broader biological effects remain insufficiently characterized. Notably, wild-type and male APP/PS1 mice do not show similar changes upon treatment. This raises the question of whether the observed effects are specific to enzyme inhibition or secondary to altered systemic palmitate levels. The authors should discuss or test whether the observed rescue of PSD-95 palmitoylation is mechanistically linked to PPT1 inhibition, or whether other pathways might contribute. Additionally, it is plausible that other palmitoylated proteins are also affected by Palmostatin B. This is particularly important since widespread increases in protein palmitoylation have been implicated in neurotoxicity, as seen in PPT1 deficiency disorders. Moreover, elevated palmitate levels (e.g., from high-fat diets) are known to impair cognitive function. These caveats should be discussed, especially in light of the fact that proteins such as GluA1 (AMPA subunit), which require dynamic palmitoylation for synaptic plasticity, were not evaluated in this study.

Because Palm B has only limited affinity for PPT1, and given the low dose of Palm B used in our study, we don't expect significant off-target effects related to PPT1 inhibition. We have now added this clarification at lines 547-549.

Minor Comments

In the dendritic spine analysis, while the number of analyzed dendritic segments is specified, the number of biological replicates (animals) is not. This information is essential for assessing statistical power. For spine data where multiple segments come from the same animal, nested ANOVA is the appropriate statistical method and should be applied or justified otherwise.

We thank the reviewer for this suggestion, we did perform a nested ANOVA on the spine analysis data, see Methods and figure legends for details.

We thank the reviewers for reviewing our revised manuscript. Please see our point-by-point answers below. For clarity, all reviewers' comments are *italicized* and our responses are not.

Reviewers' comments:

Reviewer #1 (Remarks to the Author):

The authors addressed most of my concerns, and the manuscript has been improved. However, I still have a few major concerns.

Major concerns:

1. Despite the authors' response to my previous concern, there is still concern about the specificity of the PF11 antibody used to label palmitoylated PSD-95. The original paper using the PF11 antibody (Fukata et al., 2013, JCB) shows convincing overlap between PF11 and total PSD-95. Please refer to Figure 1E of the Fukata et al., 2013, JCB paper for images. It appears that the PF11 antibody used in this study (#AG-27B-0021-C100, AdipoGen) is ineffective. Are there any publications using this PF11 antibody from AdipoGen? The authors should replace Figures 1I-1K and 2H-2K with data obtained using a functional PF11 antibody. The current PF11 images are not suitable for publication.

We understand the concern of the reviewer with the PF11 antibody, it is not perfect. Apart from the original study mentioned (Fukata et al., 2013, JCB), there is no other images of PF11 staining in brain tissue published in the literature. However, for the reasons listed below, we are convinced that the PF11 staining presented in this manuscript is valid and think experiments using this antibody should remain in Figure 1 and 2.

- A) As noted in the previous response letter, perhaps the most convincing proof that the PF11 antibody sold by Adipogen is valid is the fact that much lower signal is observed in slices made from PSD-95 KO mice when compared to slices from WT mice (see Figure S3A).
- B) Another important point is the fact that we have been using this PF11 antibody for several years now and bought multiple vials; we get very similar results every time. Specifically, this means that the signal 1) looks very similar every time and 2) that we are consistently seeing a decrease in PF11 signal in the hippocampus of female APP/PS1 mice.
- C) To address concerns raised about this antibody by Reviewer #2, we performed an additional experiment using thin slices (8um thick), instead of the 50um slices used in other experiments. Importantly, we obtained a detailed protocol from Dr. Fukata (last author of the Fukata et al., 2013, JCB paper) for staining with PF11 in brain slices. This protocol is the same one used to obtain the data shown in Figure 1E in Fukata et al., 2013, JCB, and includes fixation of the cryostat slices in cold acetone (described in our Methods). With this protocol, we obtained only punctated signal with the PF11 antibody (see Figure S3D-E), and no signal in the cell bodies. Importantly, when we analyzed the PF11 signal intensity in female WT vs APP/PS1 mice, we confirmed that there is a significant reduction in PF11 signal in APP/PS1 mice. Therefore, the results of the other experiments using standard PFA fixation and 50um slices are valid.
- D) Reviewer #2, who was also concerned by the PF11 IHC images, was satisfied with the additional experiment mentioned in C).
- E) Other labs have used the PF11 antibody sold by Adipogen in either HEK 293 cells (Jeyifous et al, PNAS, 2016) or primary hippocampal neurons (Chowdury and Hell, Front.

Synaptic Neurosci., 2019), but not in tissue. We also performed IHC with the PF11 antibody and PSD-95 antibody in primary hippocampal neurons. In this type of preparation, we see similar overlap as in the 2019 paper.

- F) The results obtained with the PF11 antibody shown in Figures 1 and 2 are confirmed by the biochemistry assessment of PSD-95 palmitoylation using the APEGS assay. With both methods, we get to the same conclusions: PSD-95 palmitoylation is reduced in female APP/PS1 mice when compared to WT mice (not observed in males), and that Palm B injections restore PSD-95 palmitoylation in female APP/PS1 mice.

2. The experimental sequence of the Morris water maze in Figure 2A is somewhat unusual. Visible learning, a stronger stimulus for investigating egocentric learning and memory, was performed first (days 1-3). Then, the experiment progressed by gradually removing the flag (days 4-5) or making the platform invisible (days 6-7), which change the intensity and type of training. This experimental protocol differs from those described in the two referenced papers (refs #8 and #87) in the Methods section.

The rebuttal to 11-(2) is somewhat understandable. However, performing invisible learning after visible learning to confirm spatial memory complicates the interpretation, because it may reflect memory retention for egocentric learning and memory. The experimental protocol, combining invisible learning after visible learning, seems to be a more specialized method for examining egocentric memory retention or memory recall. Have there been any reports of a significant decline in memory retention or recall in female APP/PS1 mice or female AD patients? If so, this could be interpreted more convincingly. Please add this information.

From another perspective, these invisible and visible learning experiments are typically conducted separately in spatial learning research because they have different evaluation purpose. Please consider this and verify the experimental sequence and its interpretation. In addition, please add the reference from Zhang et al. (2012) to the reference section.

We thank the reviewer for the thoughtful comments about the Morris Water Maze experiments. We now realize that the cited paper from our collaborators (Reference #8, Zhang et al 2016, Rissman lab) did not use the same protocol with the visible and invisible learning phases. However, they do so in a more recent publication (<https://doi.org/10.1186/s40478-016-0410-8>). We now cite this publication and the recent publication mentioned in the previous response letter (Curdt et al. Scientific Reports, 2022) at line 726. See below a screenshot of the Method section of that paper, indicating a protocol that is very similar to the one we used for the Morris Water Maze. We apologize for not including this citation in the previous version of the paper. As for the Zhang et al 2012 citation, we meant Zhang et al 2016 and apologize for that mistake.

Morris water maze. To evaluate spatial reference memory, the Morris Water Maze (MWM) was performed as previously described^{10,15}. Therefore, mice were trained to search a hidden circular platform (10 cm) in a pool

ficreports/

(110 cm diameter) filled with opaque water by using different cues. The pool was filled with tap water mixed with non-toxic white paint. During the whole test the water temperature was maintained at 20 ± 2 °C. The pool was divided into four virtual quadrants based on the platform localization: left, right, opposite, and target.

Each mouse went through 3 days of cued training, 5 days of acquisition training and a final probe trial. During the cued training the platform was marked with a triangular flag. During the cued training no additional distal cues were present. Mice were introduced into the water near the edge of the pool facing the wall. They were given one minute to find the submerged platform. The platform was located in the upper third of the quadrant (14 cm from the rim of the pool). If they failed to find the platform in one minute, they were gently guided to it. Every mouse had to sit on the platform for 10 s before being removed from the pool. To prevent hypothermia mice were kept in front of a heat lamp until they were dry. The cued training consisted of four trials per day with an average inter-trial interval of 15 min. The start position and the position of the platform changed for every trial.

Forty-eight hours after the last cued training day, 5 days of acquisition training started. For the acquisition training the flag was removed from the platform. In addition to the distal cues in the room, proximal visual cues were attached to the edge of the pool. During the acquisition training, mice were placed in the water from one of four predefined entry points, while the location of the platform remained stationary. Trials were performed as during the cued training phase.

Twenty-four hours after the last acquisition trial, a probe trial was performed to assess spatial reference memory. For the probe trial the platform was removed from the pool, and mice were introduced into the water

3. The authors provided new data showing the palmitoylation of total proteins in Figures 1A and 1B. Please expand the upper immunoblot image in Figure 1B to include the proteins around 245 kDa, as shown in Figure 1A.

We thank the reviewer for pointing that out. It is now fixed.

Minor concerns:

1. The (V) and (PB) labels in Figures 2H and 2I are incorrect. The group labels should be on the left side of the images. Please correct them.

2. In the Y-axis of Figure 2J, (N to WT) may need to be corrected to (N to V).

We thank the reviewer for pointing that out. It is now fixed.

Reviewer #2 (Remarks to the Author):

The authors addressed all the Reviewer's concerns.

Reviewer #3 (Remarks to the Author):

I co-reviewed this manuscript with one of the reviewers who provided the listed reports. This is part of the Communications Biology initiative to facilitate training in peer review and to provide appropriate recognition for Early Career Researchers who co-review manuscripts.

Reviewer #4 (Remarks to the Author):

I have reviewed the revised manuscript, and I am pleased with the authors' thorough and thoughtful responses to my previous comments. All concerns have been fully and satisfactorily addressed, and the revisions have significantly improved the clarity and overall quality of the work.

I have no further suggestions for the authors at this stage. The manuscript is well written, scientifically sound, and suitable for publication in its current form.

I therefore recommend acceptance of the manuscript.

We thank the reviewer for their careful re-assessment of our revised manuscript and for their kind words.

We thank the reviewer for reviewing our revised manuscript. Please see our point-by-point answers below. For clarity, all reviewers' comments are *italicized* and our responses are not.

Reviewers' comments:

Reviewer #1 (Remarks to the Author):

The authors responded to the points raised in the second revision, but some issues remain unresolved.

1. Regarding the concerns about the specificity of the PF11 antibody, the ICC images provided by the authors in their response 1-E further confirm that the PF11 antibody used in this study is not specific to palmitoylated PSD-95.

In response 1-E, the images show that the antibody used in Chowdury and Hell (2019) worked well in ICC, but the antibody used in this study did not work, even in ICC. Since the Chowdury and Hell (2019) study used the PF11 antibody obtained from the Fukata lab, I recommend that the authors obtain the PF11 antibody from the Fukata lab and replace the Figures 1I-1K and 2H-2K. In my opinion, AdipoGen's PF11 antibody is not suitable for IHC or ICC, whereas the PF11 antibody from the Fukata lab works for both.

I agree with the author's main point that palmitoylated PSD-95 levels are lower in the hippocampus of female APP/PS1 mice. However, the data presented in Figures 1I-1K and 2H-2K, which were labeled by AdipoGen's PF11 antibody, are unreliable.

We understand the concerns of Reviewer with the PF11 antibody but find the suggestion of requesting the antibody from the Fukata lab and repeat a significant portion of the experiments in the manuscript unjustified.

Assuming that the Fukata lab can provide this antibody to us in a timely manner, these experiments would require at least 6 months to complete and a significant number (20-50) of additional 9-10 months old WT and APP/PS1 mice.

Importantly, the reviewer mentions that: " I agree with the author's main point that palmitoylated PSD-95 levels are lower in the hippocampus of female APP/PS1 mice"; therefore, these extensive supplemental experiments are unlikely to yield a meaningful outcome. Instead of spending considerable time and resources on the requested experiments, we decided to directly acknowledge this limitation of our study. To do so:

- We removed all IHC images from the main figures and instead included these data in the supplementary information. This way, the biochemical characterization of PSD-95 palmitoylation are treated as the main findings.
- We removed claims of specificity and mentions of using the PF11 to 'confirm' the biochemical results. See lines 159 and 174-175.
- Each time the PF11 antibody was mentioned, we specified that we are using the commercial version of the antibody and not the one produced by the Fukata lab. See lines 161, 171, 297 and the titles and figure legends of Figure S4, S5, S6 and S9.
- We added text in the discussion to clearly address the issues with the commercial PF11 antibody, lines 635-643.
- Copied here as well: Another limitation of our study is related to the use of the commercially available PF11 antibody. To our knowledge, all other published immunofluorescence images in neurons or brain slices were obtained with the PF11

antibody made by the Fukata laboratory^{24,31,46}. We acknowledge that there is less overlap between PSD-95 and PF11 signals in our images, which indicates that some of the PF11 signal comes from unspecific staining and that the commercially available PF11 should be used with caution. However, as with the biochemical experiments shown in Figures 1 and 2, we consistently measured a reduction in PF11 signal intensity in the hippocampus of female APP/PS1 mice (Fig. S4) and a rescue of the PF11 signal in Palm B injected mice (Fig. S9).

As the reviewer can see, we clearly and directly address the overlap and unspecific staining issues as well as warning the readers about using this antibody with caution.

2. Regarding the Morris water maze experiments, the replacement of the references has been appropriately addressed. However, the previous comment was not only about the references themselves, but also about the need for an appropriate interpretation of the unusual experimental protocol used in the Morris water maze test in this study. Since this protocol differs from conventional designs, a tailored interpretation is necessary. Unfortunately, this aspect has not been addressed in the revised version.

I encourage the authors to provide a clear discussion that interprets the behavioral results in the context of this modified experimental protocol. In addition, have there been previous reports indicating impaired memory recall or retention specifically in females? Including a discussion of such prior findings would provide a novel perspective and enrich the manuscript further.

In the previous response letter, we did correct the references regarding the Morris Water Maze test. Importantly, the new references provided use a protocol that is almost exactly the same as ours and include both visible and invisible learning (named cued and acquisition training). This argues that our protocol is not unusual and does not require a tailored interpretation. To make our protocol clearer and give more details about the two training phases, we added some text in the results section (see lines 253-254 and 276-277). We are cautious about overinterpreting the behavioral results and therefore did not add text in the discussion. We hope the reviewer understands.

We thank the reviewer and the Communications Biology Editorial team for reviewing our revised manuscript. Please see our point-by-point answers below. For clarity, all editors' comments are *italicized* and our responses are not.

In our last decision we asked you to contact the Fukata lab to request their antibody. After you contacted us with your concern about the value of repeating a large amount of animal work in order to repeat the immunohistology, we offered you a compromise; we requested that the APEGS data should be treated as the main assay for palmitoylated PSD-95 and given priority over IHC which relies on the AdipoGen PF11 antibody. Both your Results and Discussion should make clear the limitations of the AdipoGen PF11 antibody and that the IHC data should be treated with caution. Although we appreciate that you have moved the relevant figures to the Supplement, and have added a line about the limitation to your Discussion, we have not found any evidence of prioritisation of the APEGS data or reframing the IHC within the context of the clear limitations of the antibody. At present the IHC analysis is still reported in full and takes up twice as much space as the APEGS data (lines 144-156 vs lines 153-183). We do not feel that this response sufficiently recognises the issue that has been raised by the Reviewer and as such will offer you one more opportunity to revise your manuscript. We make the following specific requests:

We thank the editorial team for providing this compromise and apologize that our revised manuscript sent on January 9 did not meet all requirements. As can be seen below and in our revised manuscript, we took this last opportunity to revise our manuscript seriously and carefully responded to all requests.

1- Line 160: "Using a commercial version of a conformation specific antibody designed to detect palmitoylated PSD-95, PF11 [46]" unless you are able to provide citable evidence that these are the same antibody you must not conflate the Fukata antibody with the antibody supplied by AdipoGen (#AG-27B-0021-C100). Please carefully check your manuscript to remove any such statements.

To make sure that there is no confusion between the Fukata antibody and the AdipoGen antibody, we revised the text at line 160 and removed the citation of the Fukata paper. This paper is now mentioned only in the Methods, see line 757 and citation # 85.

*2- Line 164: Please add the statement flanked by**:*

*IHC conditions were optimized using PSD-95 knock-out mice to reduce unspecific binding with the PSD-95 and PF11 antibodies**, however, even after optimisation, residual staining is apparent in the PSD-95 KO when probed with the PF11 antibody (Fig. S3A). It is also important to note that there is a lack of co-localisation between the PSD-95 antibody and the AdipoGen PF11 antibody (Fig. S3a, S4a, b & d, S5a, b, c & d, S6a & b, S9a & b). These data suggest that the AdipoGen PF11 antibody produces non-specific signal. As such these figures are included only as Supplementary Figures. They may be considered indicative, but should not be interpreted as confirmatory of the APEGS data.***

This statement is now added. See lines 165-171.

3- When describing the results of IHC using the PF11 antibody, please refer only to “PF11 signal” rather than “PSD-95 palmitoylation”.

This has been corrected, both in the main text and in the Supplementary information.

4- Lines 167-183: Please rewrite to deemphasise and summarise the findings, being clear, as above, about the strength of conclusions that should be drawn from these data. Line 245-249: Please rewrite to deemphasise and summarise the findings, being clear, as above, about the strength of conclusions that should be drawn from these data.

For the IHC data related to Figure 1: to de-emphasize the IHC findings, we removed the quantification data, placed the description of the thin slice IHC in the Methods section and summarized the findings related to Figure S5. We also put the text describing Figure S6 in a separate paragraph and changed the order of the panels to emphasize the Western Blotting results (See lines 179-184). With these changes, there is more text describing the biochemical results (lines 141-156 and lines 179-181 = 19 lines total) than the IHC results (lines 158-165 (6 lines total, bug with line counting in Word), lines 172-177 and line 182: 13 lines total (not including the statement about the issues with the PF11 antibody), if we include this statement, the number of lines describing the APEGS assay results and the IHC results is the same: 19 lines). We therefore significantly reduced the emphasis on the IHC results and importantly informed the readers about the issues with the commercial PF11 antibody.

For the IHC data related to Figure 2: we also removed the IHC quantification data and summarized findings (see lines 304-305). We also removed the IHC data from Figure S11 as these were not required (see point 5 below).

5- Figure S11 e and f. If these PF11 IHC data are not described in the manuscript then please remove the Figure panels.

We agree with the editorial team, these figure panels are now removed.

6- Please also be sure to accurately describe your data, for example on line 171 a statement is made related to Fig. S4a-c about “both sexes” while the figure and its legend refer to female mice only.

Thank you for this note. Upon verifying this statement and the related figure, we revised the text at line 174. The example images shown in Figure S4A,B are for female only, but the quantification data shown in Figure S4C is for both males and females.

As requested, we also took this opportunity to carefully reassess our writing to make sure all data is described accurately.

We thank the reviewer and the Communications Biology Editorial team for reviewing our revised manuscript. Please see our point-by-point answers below. For clarity, all reviewers' comments are *italicized* and our responses are not.

REVIEWERS' COMMENTS:

Reviewer #1 (Remarks to the Author):

I would like to thank the editorial team for their great compromise.

Regarding line 702 (or 761 in the marked-up version), where the authors refer to the Fukata paper "as in the original publication" in relation to their IHC optimization, I would suggest removing "as in the original publication," since Fukata's and AdipoGen's antibodies, despite having the same name, are different.

The following should be sufficient:

"we performed an experiment using 8 μ m thin sections fixed in acetone⁸⁴"

Yes, we agree with the reviewer about this point. This has been corrected. See lines 566-567 (or 682-683 in the marked up version).